# TRAINING LANGUAGE MODELS TO CRITIQUE WITH MULTI-AGENT FEEDBACK

## ABSTRACT

Critique ability, a meta-cognitive capability of humans, presents significant challenges for LLMs to improve. Recent works primarily rely on supervised fine-tuning (SFT) using critiques generated by a single LLM like GPT-4. However, these model-generated critiques often exhibit flaws due to the inherent complexity of the critique. Consequently, fine-tuning LLMs on such flawed critiques typically limits the model's performance and propagates these flaws into the learned model. To overcome these challenges, this paper proposes a novel data generation pipeline, named MultiCritique, that improves the critique ability of LLMs by utilizing multi-agent feedback in both the SFT and reinforcement learning (RL) stages. First, our data generation pipeline aggregates high-quality critiques from multiple agents instead of a single model, with crucial information as input for simplifying the critique. Furthermore, our pipeline improves the preference accuracy of critique quality through multi-agent feedback, facilitating the effectiveness of RL in improving the critique ability of LLMs. Based on our proposed MultiCritique data generation pipeline, we construct the MultiCritiqueDataset for the SFT and RL fine-tuning stages. Extensive experimental results on two benchmarks demonstrate: 1) the superior quality of our constructed SFT dataset compared to existing critique datasets; 2) additional improvements to the critique ability of LLMs brought by the RL stage. Notably, our fine-tuned 7B model significantly surpasses other advanced 7B-13B open-source models, approaching the performance of advanced 70B LLMs and GPT-4. Codes, datasets and model weights will be publicly available.

## 1 INTRODUCTION

The critique ability, *i.e.*, the capability to identify and refine flaws in responses, has been widely used to facilitate reliable automatic evaluation and self-improvement of LLMs (Lan et al., 2024; Wu et al., 2024). As a meta-cognitive capability (Toy et al., 2024; Wang & Zhao, 2024), critique ability requires LLMs to possess a deep understanding of user queries and evaluated responses beyond mere criticism (Kim et al., 2024; Zheng et al., 2023). Therefore, it is challenging to improve the critique ability of LLMs (Lan et al., 2024; Lin et al., 2024).

Recent researches indicate that open-source LLMs usually demonstrate weaker critique abilities compared to advanced closed-source models (Lan et al., 2024; Lin et al., 2024). Therefore, numerous works have been proposed to improve the critique ability of LLMs, primarily through Supervised Fine-Tuning (SFT) using critiques generated by one single LLM, typically GPT-4 (Li et al., 2024b; Cui et al., 2023). However, these model-generated critiques often exhibit flaws like inaccurate descriptions or suggestions about flaws in responses due to the complexity of the critique (Verga et al., 2024; Lan et al., 2024; Liu et al., 2024d). Consequently, the fine-tuned LLMs on these datasets are constrained by these flawed critiques, and these flaws in critiques are propagated into the learned models during the SFT stage, leading to the potential amplification of these issues.

To address these challenges, in this paper, we develop a novel data generation pipeline named **MultiCritique**. This approach utilizes multi-agent feedback to construct the high-quality dataset for the SFT and RL fine-tuning stages. First of all, to mitigate the effects of flawed critiques generated by one single LLM, we develop the **MultiCritique-SFT** pipeline, as shown in Figure 1 (Step 2). Specifically, it first collects the multi-agent analytical critiques from four advanced LLMs rather

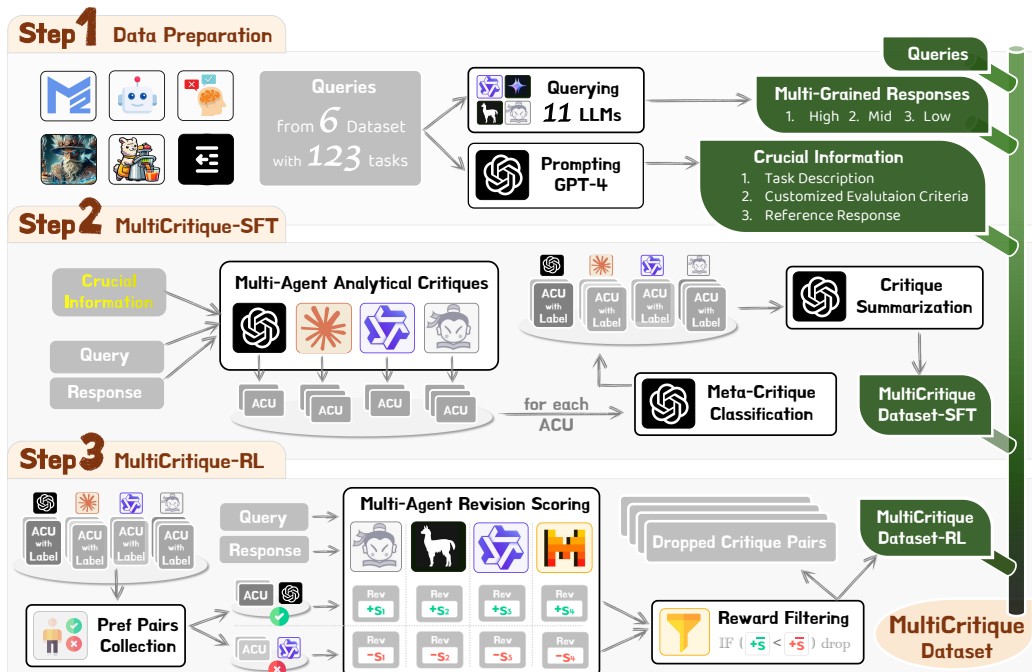

Figure 1: The overview of our proposed MultiCritique data generation pipeline. First, we prepare queries and evaluate responses and crucial information (Step 1). Then, we conduct the MultiCritique-SFT pipeline to construct the high-quality SFT critique dataset (Step 2). Finally, we conduct the MultiCritique-RL pipeline to construct the preference critique dataset for the RL stage (Step 3). An ACU is a structured unit for identifying one specific flaw in the evaluated response. A list of model-generated ACUs denotes the analytical critique.

than one single LLM, with three crucial information as input for simplifying the critique: (1) A detailed task description of the user query; (2) A customized two-tier structured evaluation criteria; and (3) A reference response tailored to satisfy these criteria. Each model-generated analytical critique consists of a list of **A**nalytical **C**ritique **U**nits (ACUs), with each ACU identifying one specific flaw in the evaluated responses. Then, GPT-4 conducts the meta-critique classification process to classify these ACUs into seven quality categories automatically (Lan et al., 2024), given all multi-agent analytical critiques as context. The process concludes by summarizing a final analytical critique that aggregates high-quality ACUs while discarding flawed ones, effectively combining the advantages of multiple models.

Second, to move beyond the mere behavior cloning on model-generated critiques in the SFT dataset, we develop the **MultiCritique-RL** data generation pipeline to construct the high-quality preference critique dataset via multi-agent feedback, facilitating the effectiveness of RL in improving the critique ability of LLMs. Specifically, as shown in Figure 1 (Step 3), chosen (**+**) and rejected (**-**) analytical critiques are naturally paired via meta-critique classification in the MultiCritique-SFT pipeline, where chosen analytical critiques exhibit fewer and minor flawed ACUs than rejected ones. However, the preference between these chosen and rejected analytical critiques might be inaccurate due to the complex meta-critique analysis and limited model capability (Lan et al., 2024; Sun et al., 2024). To address this issue, we then propose the **M**ulti-**A**gent-**R**evision-**S**coring (MARS) filtering to refine the preference critique dataset, ensuring the chosen analytical critiques lead to superior revisions compared to rejected ones across multiple models. These preference pairs are used to fine-tune a reward model, guiding the RL process to improve critique ability further. Unlike recent works that utilize preference learning to improve the critique ability of LLMs, such as Themis (Hu et al., 2024) and CriticGPT (McAleese et al., 2024), our proposed MultiCritique-RL pipeline does not require any human annotations, thereby demonstrating better scalability.

Based on our proposed MultiCritique data generation pipeline, we construct the **MultiCritique-Dataset** for improving critique ability, consisting of two splits for the SFT and RL fine-tuning stages (right part of Figure 1). Extensive experiments on two benchmarks demonstrate the supe-

rior quality of our proposed MultiCritiqueDataset compared to other critique datasets. Specifically, even if the scales of existing datasets are over three and eight times larger than our proposed dataset, the model fine-tuned during the SFT stage on our proposed MultiCritiqueDataset still significantly outperforms those fine-tuned on other datasets, with 21.48% and 22.50% average performance gain on CRITICEVAL (Lan et al., 2024) and CRITICBENCH (Lin et al., 2024) benchmarks, respectively. Furthermore, the RL fine-tuning stage on MultiCritiqueDataset further boosts the critique ability of the SFT model, resulting in 6.3% and 0.51% absolute improvements on CRITICEVAL and CRIT-ICBENCH. Besides, ablation studies confirm the positive contributions of crucial information in simplifying the critique and the roles of our proposed MultiCritique data generation pipeline in enhancing dataset quality. In summary, the InternLM2-7B-Chat-SFT model that fine-tuned by SFT and RL stages on our proposed MultiCritiqueDataset, significantly outperforms advanced 7B-13B instruction-tuned and critique-tuned LLMs as well as the GPT-3.5-Turbo model, approaching advanced 70B LLMs and GPT-4. For example, our model approaches GPT-4 on CRITICBENCH, with an F1 score of 75.66% vs. 78.75%, significantly outperforming average performance of 7B-13B instruction-tuned and critique-tuned LLMs (55.58%).

## 2 RELATED WORK

**Critique Ability of LLMs** The critique ability of LLMs has been applied in three key areas: (1) Reliable Automatic Evaluation: LLMs can achieve high correlation with human annotators in response evaluation (Liu et al., 2023; Zheng et al., 2023; Saunders et al., 2022); (2) Self-Improvement of LLMs: This ability facilitates the self-improvement of LLMs during inference and training stages (Yuan et al., 2024; Wu et al., 2024); (3) Robust Reward Modeling: Textual critiques contribute to robust reward modeling by providing detailed feedback (Ye et al., 2024; Zhang et al., 2024b). Recently, to bridge the gap between the weak open-source LLMs and advanced closed-source models, various datasets have been developed to enhance the critique capability of LLMs by mimicking GPT-4 behavior (Lan et al., 2024; Lin et al., 2024). For example, datasets such as UltraFeedback (Cui et al., 2023), Auto-J (Li et al., 2024b) and Prometheus (Kim et al., 2024), collect critiques from GPT-4 to fine-tune open-source LLMs like Llama2-13B. Despite their potential, these datasets often suffer from quality issues due to the inherent complexity of critiques (Verga et al., 2024). Compared with these existing works, we first propose a novel data generation pipeline, MultiCritique, to collect high-quality critiques from multi-agents. Then, reinforcement learning is used to improve the critique ability of LLMs, moving beyond the mere behavior cloning on the supervised critique dataset.

**Preference-based Reinforcement Learning (RL)** Reinforcement learning (RL) algorithms, like PPO (Schulman et al., 2017), are widely utilized to guide LLMs to generate responses that are more preferred by humans (Yang et al., 2024b; Xu et al., 2024; Yuan et al., 2024). RL algorithms typically employ a reward model as a proxy for human judgment, learning through human-annotated pairwise comparison of responses. This is often called Reinforcement Learning from Human Feedback (RLHF) (Stiennon et al., 2022; Ouyang et al., 2022). Recently, some works have been proposed to improve the critique ability through reinforcement learning. For example, CriticGPT (McAleese et al., 2024) improves the critique ability of ChatGPT through RLHF. Similarly, Themis (Hu et al., 2024), Self-Taught Evaluator (Wang et al., 2024b), SFR (Wang et al., 2024a) collect the preference critique pairs by examining consistency between the LLM's judgment scores and the human-annotated judgment scores. However, their solution is limited due to the high cost of human annotations. In contrast, our proposed MultiCritique pipeline automatically collects preference critique datasets via a multi-agent framework without human supervision. Our concurrent work, Meta-Rewarding (Wu et al., 2024), collects preference critique pairs using optimized LLM as the meta-judge model. Compared with Meta-Rewarding, we propose the Multi-Agent-Revision-Scoring (MARS) to filter high-quality critique pairs by using the quality of their revisions as an indicator of critique quality, thereby facilitating a more robust RL.

**Multi-agent Framework** Current multi-agent frameworks are widely used in two applications: (1) multi-agent framework for LLM alignment; and (2) multi-agent framework for LLM-based evaluation. In the domain of LLM alignment, extensive research, including multi-agent debate frameworks (Du et al., 2023; Khan et al., 2024), have proven the effectiveness of multi-agents in enhancing LLM's alignment and response quality through fostering more divergent thinking and aggregating the diverse opinions of multiple LLMs (Liang et al., 2024; Zhang et al., 2024a; Ji et al.,

2024). Notable examples include Arena Learning (Luo et al., 2024), which leverages a Llama-3-70B model (Meta, 2024) as a judge to evaluate battle results between multiple models for target LLM improvement. Similarly, Stable Alignment (Liu et al., 2024a) employs multi-agent critiques for alignment refinement. While these works primarily address LLM alignment, our work proposes a unified multi-agent feedback framework that improves the critique capabilities of LLMs from the SFT and RL phases.

In the context of critique, recent multi-agent frameworks improve the reliability of LLM-based evaluation during the inference stage. For example, ChatEval (Chan et al., 2023) enhances the reliability of LLM-based automatic evaluations via multi-agent debate. PoLL (Verga et al., 2024) addresses the model-specific biases by pooling judgment scores of a panel of small models. Unlike these works, our work proposes a multi-agent feedback framework to improve the critique ability of LLMs through the SFT and RL fine-tuning stages.

## 3 METHOD

### 3.1 DATA PREPARATION

As shown in Figure 1 (Step 1), we collect diverse queries and evaluated responses before running our proposed MultiCritique data generation pipeline. Besides, we also elicit crucial information from the queries to simplify the critique. Details are elaborated as follows.

**Diverse Queries Collection** We first compile queries from several well-established datasets. These include alignment datasets such as OpenHermes-2.5, DEITA (Liu et al., 2024b) and OpenAssistant (Köpf et al., 2023); mathematical and coding reasoning datasets like MetaMathQA (Yu et al., 2024) and CodeFeedback (Zheng et al., 2024); as well as the critique dataset Auto-J (Li et al., 2024b). In total, we sample 10.7K queries covering 123 diverse task scenarios. The details of these 123 tasks are listed in Table 12 in Appendix D.

**Diverse Evaluated Responses Collection** Once the queries are collected, we use eleven LLMs with different capabilities to generate responses with varying response qualities. These responses are then evaluated using the robust reward model—InternLM2-20B-reward (Cai et al., 2024), which performs well in RewardBench (Lambert et al., 2024). We select three low-, medium- and high-quality responses for each query, ensuring a significant performance gap. In total, 10.7K×3=32.1K query-response pairs are collected. Due to the page limitation, the implementation details and statistical of these response quality are placed in Appendix C.1.

**Crucial Information Collection** Once the query-response pairs are collected, we elicit three crucial information sequentially using GPT-4 to simplify the critique: **(1) Task Description**: Our preliminary findings indicate that LLMs may misunderstand the objectives of queries. By prompting GPT-4 to describe the task, we can mitigate this issue to some extent; **(2) Customized Evaluation Criteria**: Once the task description is obtained, we propose generating customized two-tier structure evaluation criteria tailored to each query to guide effective critiques. The first tier outlines the fundamental evaluation dimensions for the task, while the second tier offers more customized criteria. Each criterion is structured with a name, description and level of importance (Li et al., 2024b); **(3) Reference Response**: Finally, we generate reference responses that satisfy all customized evaluation criteria. Since reference responses tend to produce critiques that lack diversity for mathematical and coding questions, we set reference answers as empty for these two tasks.

Following these three steps, we collect $\mathbb{N}$=32.1K samples $\{(q_i, r_i, \mathcal{CI}_i)\}_{i=1}^{\mathbb{N}}$, where $q_i, r_i, \mathcal{CI}_i$ represents the $i$-th query, evaluated response and corresponding crucial information in the dataset. Please refer to Appendix C and Appendix D for more details and statistical information about the collection of queries and evaluated responses.

### 3.2 MULTICRITIQUE-SFT DATA GENERATION PIPELINE

After collecting query-response and crucial information, we introduce our proposed MultiCritique-SFT data generation pipeline to construct the high-quality SFT dataset. The overview of MultiCritique-SFT pipeline is shown in Figure 1 (Step 2). This pipeline gathers multi-agent critiques to mitigate flawed critiques generated by one single LLM.

To achieve this goal, we aim to aggregate the high-quality content in multi-agent critiques into a comprehensive and accurate critique. Therefore, we first collect detailed analytical critiques simultaneously from multiple LLMs (**Multi-Agent Analytical Critique**). Each analytical critiques consist of a list of **A**nalytical **C**ritique **U**nits (ACUs). Then, the GPT-4 classifies each ACU into one of seven quality categories (**Meta-Critique Classification**). Finally, a final critique is aggregated by summarizing the accurate ACU while discarding the flawed ones (**Critique Summarization**). Unlike previous works allowing multi-agent discussions or debates (Chan et al., 2023), our preliminary study observes that LLM's critiques could be easily influenced by other critiques, potentially reducing the diversity and comprehensiveness of critiques. Thus, we do not allow the inner-model discussions. Instead, we utilize GPT-4 as a summarizer to evaluate each critique content given all LLMs critiques as context, since previous works (Lan et al., 2024; Lin et al., 2024) have proven that only GPT-4 could effectively conduct the meta-critique.

**Multi-agent Analytical Critiques** We employ four LLMs to simultaneously critique the evaluated responses: GPT-4, Claude-1-instant, Qwen-1.5-72B-Chat and InternLM2-20B-Chat, which exhibits strong performance on CRITICEVAL benchmark (Lan et al., 2024). Each LLM performs sentence-by-sentence and cross-sentence critique and generates a structured analytical critique, which is a list of **A**nalytical **C**ritique **U**nits (ACUs). An ACU is a structured unit for identifying and addressing a specific flaw in the evaluated response, consisting of five values about the flaw: (1) the location[1]; (2) the description; (3) suggestions for revision; (4) the criteria type; and (5) the severity. The structured ACUs not only benefits the robust subsequent meta-critique process (Sun et al., 2024) but also exhibit great interprebability.

**Meta-Critique Classification** Since LLMs often produce flawed critiques due to the limited model capability (Lan et al., 2024; Wang et al., 2023), it is crucial to identify these flaws before aggregating them into a final comprehensive and accurate analytical critique. To achieve this goal, we employ GPT-4 to judge the quality of these model-generated analytical critiques, a concept known as meta-critique (Sun et al., 2024; Lan et al., 2024). Specifically, GPT-4 classifies each ACU into one of seven quality categories by considering all the multi-agent analytical critiques in the context. These quality categories are determined by human annotators and are associated with human-annotated severity scores ranging from 1 to 5. Each quality category reflects one specific flaw in an ACU. Please refer to Appendix J.5 for more details. Importantly, for one model-generated analytical critique, the accumulated severity scores of its ACUs could indicate its quality, whereas a higher accumulated severity score indicates lower analytical critique quality.

**Critique Summarization** After the previous steps, GPT-4 summarizes these ACUs into a comprehensive analytical critique. This process involves retaining and merging accurate ACUs generated by multiple LLMs while modifying or excluding those identified as flawed. Except for the final summarized analytical critique, we also prompt GPT-4 to generate an overall description and judgment score for the evaluated response. Finally, the final analytical critique, description and judgment score are concatenated as the final critique for the evaluated response, denoted as $\mathcal{C}$.

By following previous steps of MultiCritique-SFT, we construct a supervised fine-tuning dataset MultiCritiqueDataset-SFT, consisting of $\mathbb{N}$=32.1K samples: $\{(q_i, r_i, \mathcal{CI}_i, \mathcal{C}_i)\}_{i=1}^{\mathbb{N}}$.

### 3.3 MULTICRITIQUE-RL DATA GENERATION PIPELINE

Beyond the behavior cloning on the supervised dataset, we also conduct the MultiCritique-RL data generation pipeline to construct the preference critique dataset for training reward models. As shown in Figure 1 (Step 3), the MultiCritique-RL pipeline involves two steps.

**Preference Pairs Collection** Four model-generated analytical critiques could be naturally collected from our proposed MultiCritique-SFT data generation pipeline. As described above, the quality of ACUs in each analytical critique is classified by meta-critique, and each model-generated analytical critique is associated with an accumulated severity score. Except for four model-generated analytical critiques, we also supply the summarized final analytical critique with its accumulated severity score of 0. For $i$-th sample in the SFT dataset, the chosen and rejected critiques, denoted as $C_{i,j_+}, C_{i,j_-}$, are paired when there exists a significant performance gap between them, *i.e.,* the difference in their accumulated severity scores is greater than the threshold, which we determined to be 5. It should

---

[1]We introduce a pre-processing step to label sentences in evaluated responses, as detailed in Appendix C.1.

be noted that the descriptions and final judgment score of response quality are primarily based on the final summarized analytical critique. Since they are easy instruction-following tasks, we do not collect their preference pairs for the RL fine-tuning stage.

**Multi-Agent-Revision-Scoring (MARS) Filtering** The meta-critique analyses might be inaccurate due to the complex meta-critiques and limited GPT-4 capability (Lan et al., 2024), leading to the noise in the preference dataset. To address this issue, we leverage the quality of revision as an indicator of critique quality, and propose the Multi-Agent-Revision-Scoring (MARS) filtering to refine the preference dataset. Specifically, four 7B LLMs first revise the evaluated response based on each critique, each performing eight revisions, resulting in a total of 4×8=32 revisions. The reason for using multiple LLMs rather than one single model for revisions is to ensure the reliability and robustness of the evaluation. These revisions are then evaluated using the InternLM2-20B-reward (Cai et al., 2024). Finally, critique pairs in the preference dataset are reserved if the chosen critique's average reward score is higher than the rejected critique's score. For mathematical problems, we compute the exact answer matching rather than reward model scores.

In summary, the preference dataset MultiCritiqueDataset-RL for the RL fine-tuning stage is constructed, consisting of $\mathbb{M}$=19.7K samples: $\{(q_i, r_i, \mathcal{CI}_i, C_{i,j_+}, C_{i,j_-})\}_{i=1}^{\mathbb{M}}$.

## 4 EXPERIMENTAL SETUP

### 4.1 IMPLEMENTATION DETAILS

Fine-tuning on MultiCritiqueDataset consists of two sequential stages:

**SFT Stage** To ensure a deep understanding of the critiques, we optimize the LLMs using the concatenation of the crucial information $\mathcal{CI}_i$ and final critiques $\mathcal{C}_i$ by minimizing Maximum Likelihood Estimation (MLE):

$$L_{\text{MLE}} = -\frac{1}{\mathbb{N}} \sum_{i=1}^{\mathbb{N}} \log p_\theta(\mathcal{CI}_i, \mathcal{C}_i | q_i, r_i) \tag{1}$$

**RL Stage** After the SFT fine-tuning stage, a reward model is first trained to classify chosen and rejected analytical critiques $C_{i,j_+}, C_{i,j_-}$ by optimizing the focal ranking loss, following previous works (Cai et al., 2024). Then, the SFT model is optimized by PPO (Schulman et al., 2017) algorithm to generate the analytical critiques with fewer flaws, guided by this reward model.

Please refer to Appendix C for implementation details about fine-tuning stages.

### 4.2 BENCHMARKS AND EVALUATION METRICS

We utilize CRITICEVAL (Lan et al., 2024) and CRITICBENCH (Lin et al., 2024) benchmarks to evaluate the critique ability of LLMs.

CRITICEVAL evaluates critique ability across nine tasks, covering alignment, common NLP and reasoning capabilities. We first evaluate the critique quality: **(1) The objective feedback evaluation** ($F_{\text{obj.}}$) calculates the Spearman correlation between LLM and human judgments on response quality; **(2) The subjective feedback evaluation** ($F_{\text{sub.}}$) involves GPT-4 assessing the textual critiques quality. These scores range from 1 to 10.

Furthermore, we evaluate the quality of revisions generated by critiques as the indicator of critique quality: **(1) The objective revision evaluation** ($R_{\text{obj.}}$) measures the average Pass Rate of five LLMs' revisions for mathematical and coding questions. CRITICEVAL evaluates the chain-of-thought (CoT) and program-of-thought (PoT) approaches for mathematics. For coding tasks, it compares two settings: with execution (CodeExec) and without execution results (CodeNE); **(2) The subjective revision evaluation** ($R_{\text{sub.}}$) is assessed by GPT-4, with scores ranging from 1 to 10.

Importantly, CRITICEVAL has proven a strong correlation between GPT-4 and humans in subjective evaluation, given the human-annotated critiques and revisions as references.

| Models | CRITICEVAL | | | | CRITICBENCH | | | | | |
|---|---|---|---|---|---|---|---|---|---|---|
| | $F_{\text{obj.}}$ | $F_{\text{sub.}}$ | $R_{\text{obj.}}$ | $R_{\text{sub.}}$ | Math | Comm. | Symb. | Algo. | Code | Overall |
| *Closed-source LLM* | | | | | | | | | | |
| **GPT-3.5-Turbo** | 61.47 | 5.06 | 15.54 | 6.20 | - | - | - | - | - | 51.44 |
| **GPT-4-Turbo** | 76.09 | 7.90 | 26.88 | 7.71 | - | - | - | - | - | 78.75 |
| *70B instruction-tuned LLMs* | | | | | | | | | | |
| **Qwen2-72B-Instruct** | 75.44 | 7.83 | 23.89 | 7.21 | 82.15 | 59.64 | 78.22 | 51.35 | 85.81 | 75.86 |
| **Llama3-70B-Instruct** | 73.28 | 7.05 | 21.97 | 6.90 | 82.35 | 60.22 | 86.31 | 54.90 | 86.16 | 76.80 |
| *7B-13B instruction-tuned and critique-tuned LLMs* | | | | | | | | | | |
| **Qwen2-7B-Instruct** | 50.49 | 5.47 | 16.21 | 5.42 | 52.25 | 26.20 | 29.55 | 9.35 | 70.23 | 45.66 |
| **Llama3-8B-Instruct** | 37.20 | 5.04 | 17.69 | 5.98 | 78.33 | **62.64** | **62.05** | **62.19** | 76.41 | 70.71 |
| **Themis-8B** | 38.07 | 4.07 | 14.43 | 2.63 | 53.34 | 27.35 | 33.16 | 35.64 | 44.33 | 42.57 |
| **Prometheus-7B (Ours)** | 38.06 | 2.54 | 18.78 | 4.57 | 59.43 | 54.28 | 31.98 | 22.82 | 67.07 | 54.25 |
| **TIGERScore-7B** | 0.64 | 3.24 | 12.89 | 4.36 | 66.62 | 38.21 | 44.52 | 27.34 | 52.49 | 52.83 |
| **TIGERScore-13B** | -2.31 | 3.39 | 15.45 | 4.54 | 68.91 | 45.47 | 53.04 | 42.86 | 44.13 | 56.28 |
| **UltraCM-13B** | 21.51 | 4.12 | 16.19 | 4.85 | 76.54 | 35.59 | 50.51 | 25.17 | 54.73 | 59.39 |
| **Auto-J-13B** | 36.05 | 4.21 | 17.69 | 5.62 | 80.02 | 50.64 | 53.06 | 52.06 | 75.61 | 67.41 |
| **InternLM2-7B-Chat-SFT** | 38.78 | 3.73 | 14.48 | 3.32 | 27.08 | 17.48 | 18.82 | 14.29 | 36.13 | 24.71 |
| **+ MultiCritiqueDataset-SFT** | 58.15 | 5.71 | **19.33** | 5.78 | **89.49** | **62.60** | 57.04 | 51.85 | **79.51** | 75.15 |
| **+ MultiCritiqueDataset-RL** | **63.28** | **6.07** | 19.26 | **6.33** | 89.36 | 60.56 | 61.51 | 57.76 | 79.32 | **75.66** |

Table 1: Overall experimental results. Results for GPT-3.5-Turbo and GPT-4-Turbo are from original paper (Lin et al., 2024). For 7B-13B models, the best performance is highlighted in bold. Results comparable to the best performance (no more than 0.2% performance gap) are also highlighted.

**CRITICBENCH** consists of 3,825 queries and evaluated responses for five challenging reasoning tasks: (1) mathematical reasoning; (2) commonsense reasoning; (3) symbolic reasoning; (4) algorithm reasoning; and (5) code generation. The correctness of the evaluated responses is annotated based on the ground-truth responses. The F1 score is used to evaluate whether LLMs can accurately identify the correctness of evaluated responses. Since critique-tuned LLMs cannot utilize few-shot samples, all models are tested under the zero-shot setting to ensure a fair comparison.

### 4.3 BASELINE DATASETS AND MODELS

**Baseline Datasets** Three critique datasets constructed by GPT-4 are compared : (1) Auto-J (Li et al., 2024b); (2) UltraFeedback (Cui et al., 2023) and (3) Feedback-Collection (Kim et al., 2024).

**Baseline Models** We evaluate both advanced closed-source and open-source instruction-tuned LLMs, including GPT-3.5-Turbo and GPT-4 (GPT-4-1106-preview), Llama3 and the Qwen2 (Yang et al., 2024a) series. Additionally, we assess several critique-tuned LLMs, such as Themis (Hu et al., 2024), TIGERScore (Jiang et al., 2023), Auto-J (Li et al., 2024b), UltraCM (Cui et al., 2023). Prometheus requires criteria and reference responses as inputs, which are unavailable in the two benchmarks. To address this, we fine-tune the InternLM2-7B-Chat-SFT model using our processed dataset, moving the evaluation criteria and reference responses into the output. Some critique-tuned LLMs, such as JudgeLM (Zhu et al., 2023), are excluded, and reasons are detailed in Appendix E.1.

More details about our evaluation setup can be found in Appendix C.

## 5 EXPERIMENTAL RESULTS

The overall experimental results are presented in Section 5.1. We then demonstrate the superiority of our constructed MultiCritiqueDataset by comparing it with other datasets in Section 5.2. Finally, we analyze the scaling phenomenon on these datasets in Section 5.3.

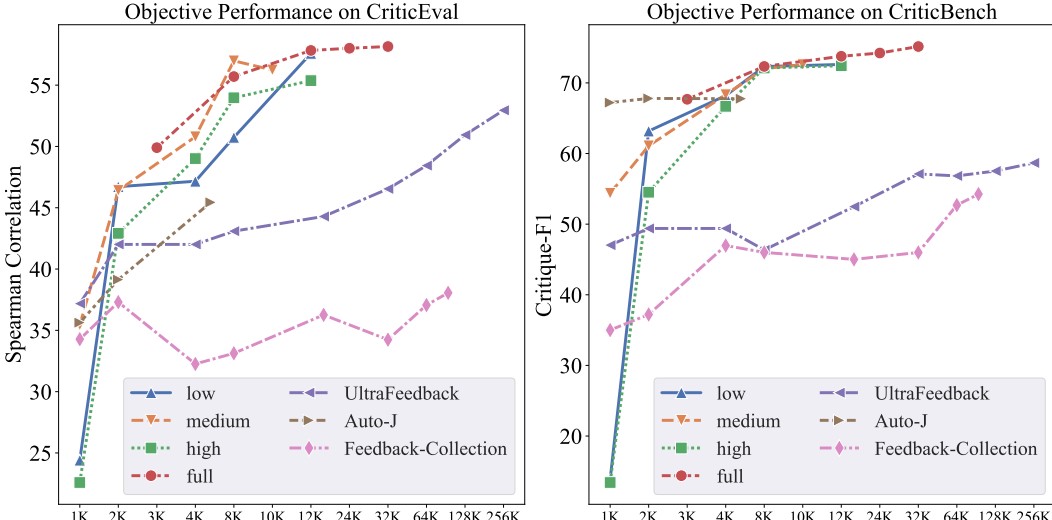

Figure 2: The correlation between the number of training samples in the SFT dataset (from 1K to 256K) and critique ability. *low, medium, high* and *full* represent the models that are trained on critiques in MultiCritiqueDataset-SFT for low-, medium-, high-quality, and all three response qualities (full), respectively.

## 5.1 OVERALL EXPERIMENTAL RESULTS

Table 1 demonstrates that both SFT and RL fine-tuning stages on our proposed MultiCritiqueDataset significantly improves the critique ability of the InternLM2-7B-Chat-SFT model. Specifically, in the CRITICEVAL subjective feedback evaluation ($F_{sub.}$), the SFT stage alone yields an absolute improvement of 19.8%, with RL stage adding an additional 6.3% absolute improvement. Consequently, our fine-tuned model not only significantly outperforms other 7B-13B instruction-tuned and critique-tuned LLMs but also uniquely surpasses GPT-3.5-Turbo model across all evaluation metrics. Furthermore, in the CRITICBENCH benchmark, our model's critique ability even approaches that of advanced 70B instruction-tuned LLMs and GPT-4, highlighting its competitive performance.

It can be observed that our model fine-tuned by RL stage has slightly lower objective revision evaluation score in CRITICEVAL for mathematical and coding tasks than the SFT model ($19.26 < 19.33$). However, when we evaluate the textual critique quality ($F_{sub.}$) for these tasks in CRITICEVAL (Table 2), RL significantly improves the critique quality across most mathematical and coding tasks. This observation suggests the inherent instability in evaluating revisions within mathematical and coding tasks.

| Models | CRITICEVAL ($F_{sub.}$) | | | |
|--------|------------|-----------|--------------|-----------|
|        | Math CoT | Math PoT | Code Exec | Code NE |
| **SFT** | 4.64 | 5.21 | 4.72 | **5.56** |
| **RL** | **5.70** | **6.21** | **4.87** | 5.33 |

Table 2: Detailed results for mathematical and coding tasks in CRITICEVAL.

## 5.2 COMPARISON WITH EXISTING CRITIQUE DATASETS

We fine-tuned the InternLM2-7B-Chat-SFT model on each dataset individually. Table 3 shows that the model fine-tuned on MultiCritique Dataset-SFT significantly outperforms those fine-tuned on other datasets, with 21.48% and 22.50% average performance gain on CRITICEVAL (subjective feedback evaluation) and CRITICBENCH, respectively. Despite the fact that the data scales of Feedback-Collection and UltraFeedback datasets are over three and eight times larger than MultiCritiqueDataset-SFT, the superior results of ours indicate its better quality.

## 5.3 SCALING PHENOMENON ON CRITIQUE-TUNED DATASETS

Figure 2 illustrates two findings: (1) The model trained on MultiCritiqueDataset-SFT consistently outperforms those trained on other datasets across most data scales, regardless of the response qual-

ity used. Specifically, Our MultiCritique framework achieves superior performance with merely 3K training samples (approximately $890 cost), surpassing baselines that require 100K-257K samples ($1,915-$3,758 in GPT-4 API costs for UltraFeedback (Cui et al., 2023) and Prometheus (Kim et al., 2024)).[2] This demonstrates a substantial improvement in data efficiency, with a factor of 2.15-4.22× over existing methods; (2) As the number of training samples increases, the critique ability of models fine-tuned with MultiCritiqueDataset-SFT improves steadily, leveling off beyond 12K samples. Besides, models utilizing critiques encompassing all response quality types (full) outperform those trained on critiques of individual quality types, indicating that diverse response qualities benefit the generalization of critique ability.

## 6 ANALYZE

In this section, we perform several ablation studies to evaluate the contributions of: (1) MultiCritique-SFT in collecting high-quality analytical critiques; (2) Crucial Information in simplifying of critiques; (3) Multi-Agent-Revision-Scoring (MARS) filtering in improving the quality of preference critiques for the RL stage, (4) MultiCritique-SFT in improving LLM's general capabilities; and (5) Generalization of MultiCritique to unseen tasks.

| SFT Models | CRITICEVAL | | | | C-BENCH |
|---|---|---|---|---|---|
| | $F_{\text{obj.}}$ | $F_{\text{sub.}}$ | $R_{\text{obj.}}$ | $R_{\text{sub.}}$ | Overall |
| **InternLM2-7B-Chat** | 38.78 | 3.73 | 14.48 | 3.32 | 24.71 |
| **+ Auto-J** | 45.44 | 3.56 | 14.63 | 3.47 | 67.76 |
| **+ UltraFeedback** | 52.95 | 4.42 | 15.81 | 3.54 | 58.67 |
| **+ Feedback-Collection** | 33.00 | 2.54 | 18.78 | 4.57 | 49.76 |
| **+ Ours** | **58.15** | **5.71** | **19.33** | **5.78** | **75.15** |

Table 3: Comparison between our constructed MultiCritiqueDataset-SFT and existing critique datasets. C-BENCH is the abbreviation of CRITICBENCH.

**Ablation Study on MultiCritique-SFT**   To examine the effectiveness of the MultiCritique-SFT pipeline, we also fine-tune the InternLM2-7B-Chat-SFT model with analytical critiques from different sources. Specifically, except for the analytical critiques generated by our proposed MultiCritique pipeline, we also collect critiques generated by each model involved in MultiCritique.[3]

| SFT Models | CRITICEVAL | | | |
|---|---|---|---|---|
| | $F_{\text{obj.}}$ | $F_{\text{sub.}}$ | $R_{\text{obj.}}$ | $R_{\text{sub.}}$ |
| **MultiCritique-SFT** | **59.74** | **5.17** | **20.92** | **6.05** |
| **GPT-4-Turbo** | 58.53 | 5.07 | 18.39 | 5.87 |
| **Claude-1-instant** | 56.77 | 5.01 | 19.00 | 5.79 |
| **Qwen-1.5-72B** | 57.30 | 4.89 | 17.74 | 5.81 |
| **InternLM2-20B** | 54.73 | 4.84 | 17.52 | 5.82 |

Table 4: Ablation study on MultiCritique-SFT.

Table 4 demonstrates that models fine-tuned with critiques generated by MultiCritique-SFT outperforms those optimized with critiques from individual models. Furthermore, there is a notable performance gap in the critiques generated by different models. For example, GPT-4 surpasses Qwen-1.5-72B-Instruct and InternLM2-20B-Chat, and all three outperforms Claude-1-instant, aligning with findings from CRITICEVAL (Lan et al., 2024).

**Ablation Study on Crucial Information**   In this section, we evaluate the contributions of three crucial information in simplifying the critiques. Specifically, we remove each crucial information during training and evaluate them in the same setting. Table 5 shows that removing each crucial information leads to a significant performance drop on most metrics in CRITICEVAL. This observation suggests that crucial information play a vital role in simplifying the critiques. Interestingly, training

| SFT Models | CRITICEVAL | | | |
|---|---|---|---|---|
| | $F_{\text{obj.}}$ | $F_{\text{sub.}}$ | $R_{\text{obj.}}$ | $R_{\text{sub.}}$ |
| **Full** | **58.15** | **5.71** | **19.33** | 5.78 |
| **- w/o Task** | 55.01 | 5.12 | 18.72 | 5.73 |
| **- w/o Criteria** | 57.28 | 5.46 | 19.12 | **6.17** |
| **- w/o Ref.** | 57.72 | 5.21 | 16.42 | 5.74 |
| **- w/o All** | 57.11 | 5.12 | 13.86 | 5.73 |

Table 5: Ablation study on crucial information.

without evaluation criteria (w/o Criteria) leads to the best performance on the subjective revision evaluation in CRITICEVAL ($R_{\text{sub.}}$). This observation suggests that while criteria benefit critiques, they might have side effects for revisions. We will explore the reasons behind this intriguing phenomenon in our future work.

---

[2]The GPT-4 API costs were calculated based on the total number of input and output tokens for both our approach and previous works.

[3]Please refer to Appendix C.2 (**Ablation Study in SFT**) for more details about the experimental setup.

**Ablation Study on MARS Filtering**  To examine the contribution of our proposed MARS pipeline, the SFT model is also fine-tuned by RL, guided by a reward model trained on the preference dataset without MARS filtering, denoted as w/o MARS. Table 6 illustrates that exclusion of MARS pipeline results in a notable decline in performance. For example, the model (w/o MARS)

| Models | CRITICEVAL | | | |
|---|---|---|---|---|
| | $F_{obj.}$ | $F_{sub.}$ | $R_{obj.}$ | $R_{sub.}$ |
| SFT Stage | 58.15 | 5.71 | **19.33** | 5.78 |
| RL Stage | **63.28** | **6.07** | 19.26 | **6.33** |
| - w/o MARS | 63.05 | 4.84 | 18.79 | 5.99 |

Table 6: Ablation study on MARS filtering.

falls short of the SFT baseline on the subjective feedback evaluation (4.84 < 5.71). This suggests that RL fine-tuning becomes instable without the MARS filtering, highlighting its contributions.

**MultiCritique Improves the General Capability of LLMs**  This section examines whether MultiCritique improves LLM's general capability. Specifically, we integrate MultiCritiqueDataset-SFT into a pool of open-source instruction-tuning datasets and optimize our base model. Then, we evaluate both the critique and general capabilities of LLMs. The evaluation of general capabilities is two-fold: (1) Objective evaluation computes the average performance on 18 famous benchmarks, like MMLU (Hendrycks et al., 2021b) and GSM8K (Cobbe et al., 2021). Please refer to the Table 10 in Appendix C.2 for the complete results; (2) Subjective evaluation uses CompassJudger toolkit (Cao et al., 2024) to evaluate performance on four general benchmarks: AlignBench (Liu et al., 2024c), AlpacaEval, Alpaca Hard (Li et al., 2023) and MTBench-101 (Bai et al., 2024). As shown in Table 7, models optimized on our integrated dataset not only maintain strong performance on critique tasks but also lead to consistent improvements in various general capabilities, *e.g.*, average 9.43% on AlpacaEval and Alpaca-Hard.

| Models | CRITICEVAL | CRITICBENCH | GENERAL BENCHMARKS | | | | |
|---|---|---|---|---|---|---|---|
| | $F_{obj.}$ | Overall | Average Objective | Align-Bench | Alpaca Eval | Alpaca Hard | MTBench 101 |
| **w/o MultiCritiqueDataset-SFT** | 41.61 | 38.80 | 55.23 | 5.07 | 23.51 | 22.78 | 7.75 |
| **w/ MultiCritiqueDataset-SFT** | **57.34** | **71.60** | **55.43** | **5.10** | **27.43** | **23.28** | **7.81** |

Table 7: Evaluate the critique and general abilities of our fine-tuned InternLM2-7B-Chat-SFT model. **Average Objective** indicates the average objective scores of LLMs over 18 benchmarks.

**Generalization to Unseen Tasks**  To further examine the generalization of our proposed MultiCritique, we exclude the training samples in the MultiCritiqueDataset-SFT for math and code tasks and evaluate the critique ability of fine-tuned InternLM2.5-7B-Chat on these two unseen tasks. As shown in Table 8, the model fine-tuned on MultiCritiqueDataset-SFT without math and code critiques still achieve significant improvements on these two tasks (88.44% > 59.46%, and 77.63% > 63.97% on math and code reasoning in CRITICBENCH), approaching the models that fine-tuned on the full-scale of MultiCritiqueDataset-SFT.

| Models | CRITICEVAL | CRITICBENCH | | | | | |
|---|---|---|---|---|---|---|---|
| | $F_{obj.}$ | Math | Comm. | Symb. | Algo. | Code | Overall |
| **InternLM2.5-7B-Chat** | 44.84 | 59.46 | 48.26 | 42.63 | 37.21 | 63.97 | 53.98 |
| **+ MultiCritiqueDataset-SFT w/o Math&Code** | **59.19** | 88.44 | 60.42 | 55.02 | 45.91 | 77.63 | 71.68 |
| **+ MultiCritiqueDataset-SFT** | 58.29 | **88.56** | **62.13** | **57.02** | **57.35** | **78.37** | **73.72** |

Table 8: Generalization evaluation of critique ability to unseen math and code reasoning tasks.

## 7  CONCLUSIONS AND FUTURE WORKS

In this paper, we propose a novel data generation pipeline, MultiCritique, to automatically construct the dataset to improve the critique ability of LLMs through SFT and RL fine-tuning stages. Extensive experiments demonstrate that MultiCritique significantly surpasses existing datasets. Additionally, the RL fine-tuning stage on MultiCritique further improves the critique abilities of LLMs. In the future, we plan to expand MultiCritique to the pairwise response comparison (Lan et al., 2024), enhancing further LLMs' ability to evaluate paired responses. Moreover, we plan to enhance the quality of MultiCritique further and tackle challenging mathematical and coding questions.

# 8 ETHICS STATEMENT

Our paper proposes a novel automatic data generation pipeline associated with a critique dataset for the SFT and RL fine-tuning stages, named MultiCritiqueDataset. As described in Appendix C, all the leveraged datasets in our work are from the publicly available datasets, which have been well processed to protect user privacy. To ensure the diversity of our datasets, we source over 32.1K user queries from 123 diverse task scenarios, and each user query consists of three evaluated responses with varying qualities, like low-, medium, and high-quality. We have proposed numerous components to ensure the quality of collected critiques, such as multi-agent revision scoring (MARS) filtering. In the future, we will continue to improve the quality and enlarge the scale of our constructed MultiCritiqueDataset. For example, we plan to extend our MultiCritique data generation pipeline to the pairwise response comparison evaluation protocol. Besides, we also plan to collect more diverse mathematical and coding queries to enhance the critique ability of fine-tuned models on these challenging reasoning tasks.

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

# A LIMITATIONS

## A.1 LIMITATIONS IN CRITIQUE PROTOCOL

So far, there are two kinds of protocols for critiques (Li et al., 2024b; Lan et al., 2024): (1) single-response evaluation; (2) pairwise response comparison. Our proposed MultiCritiqueDataset only critiques the quality of one response given the conversation history or user query, while the pairwise response comparison is not considered. Our proposed data generation pipeline could be easily modified to collect high-quality datasets for pairwise response comparison. In the future, we will extend our proposed MultiCritiqueDataset to pairwise response comparison protocols and evaluate the models under some preference benchmarks, like RewardBench (Lambert et al., 2024), PandaLM (Wang et al., 2024c) and Preference Collection (Kim et al., 2024).

## A.2 LIMITATIONS IN MULTICRITIQUE-SFT PIPELINE

This work uses four LLMs to critique the evaluated responses in our proposed MultiCritique-SFT pipeline. The capability of these four models may be sub-optimal at present. Our work began in April 2024, and these four models exhibited strong critique capabilities at that time (Lan et al., 2024). In the future, we will continue to supplement more powerful models in the MultiCritique-SFT pipeline to improve the quality of MultiCritiqueDataset, such as Llama-3.1 and OpenAI o1 series. Besides, our work only leverages four models, and more models in the MultiCritique-SFT pipeline would introduce more perspectives for critiques. Since the cost of critique and meta-critique continue to increase given more models, we only selected four models due to the limited budget. In the future, we will explore how the number of used models affects the quality of critiques in the MultiCritique-SFT pipeline.

Unlike the existing multi-agent framework Chan et al. (2023); Zhang et al. (2024c), our proposed MultiCritique-SFT does not allow the inner-model debate and discussions. The reasons are as follows: (1) The inference efficiency of inner-model debate is much lower. It not only costs a huge amount of API quota for evaluating each response but also slows down the data generation speed; (2) Our preliminary study observes that other models easily influence LLM opinions during critique generation. This may cause the multiple LLM critiques to degrade into the opinions of one or two LLMs, and the inaccurate opinions could influence other LLMs, thereby reducing the comprehensiveness and diversity of the critiques.

In contrast, our preliminary study observes that GPT-4-based meta-critique could effectively calibrate the flawed critiques with correct critiques of other LLMs. In future work, we will further address this issue to unlock the potential of multi-agent frameworks in the critique task.

## A.3 LIMITATIONS IN MULTICRITIQUE-RL PIPELINE

In MultiCritique-RL data generation pipeline, we introduce the Multi-Agent-Revision-Scoring (MARS) filtering to refine preference critique pairs obtained from the MultiCritique-SFT pipeline, using revision quality as an indicator of critique quality. Currently, revision quality is assessed by the InternLM2-20B-reward model (Cai et al., 2024), which, despite its strong performance on the RewardBench (Lambert et al., 2024) benchmarks, may not consistently reflect the true quality of responses across all task scenarios. The most reliable method would be human annotation, but it is costly and not scalable. Therefore, we adopt this trade-off approach to balance effectiveness and efficiency in measuring revision quality. In future work, we aim to enhance the accuracy of reward modeling, facilitating better MARS filtering.

## A.4 LIMITATIONS IN EVALUATION EFFICIENCY

Compared to existing critique-tuned LLMs, like Auto-J and UltraFeedback, the models fine-tuned on our proposed MultiCritiqueDataset will generate longer sequences, consisting of the task description, two-tier criteria, reference response and critiques. Therefore, the inference efficiency of our fine-tuned models may be worse than that of existing works. However, our ablation study in Section 6 reveals that crucial information is essential for more robust and accurate critiques. In future work, we will explore how to compress the crucial information and improve the inference efficiency.

## A.5 Insufficient Investigation on Advanced 70B Models

While our experiments on smaller models (7B-8B) are comprehensive (Table 1 and Table 16), our investigation of larger models ($\geq$ 70B) like Qwen2.5-72B-Instruct (Team, 2024) remains insufficient. Our preliminary experiments demonstrate the improvements brought by MultiCritique on advanced 70B models are not very significant. Our analysis suggests two primary reasons for this phenomenon: (1) These instruction-tuned 70B models have been sufficiently optimized, exhibiting very strong critique ability. **For example, Qwen2.5-72B-Instruct achieves 83.62% F1 on Crit-icBench, surpassing GPT-4's 78.75%. Besides, its objective critique performance in Crit-icEval is also superior to GPT-4 (78.72% > 76.09%).** This high level of critique performance makes further improvements challenging; (2) The current experiments utilize only MultiCritique data for optimization. We hypothesize that integrating our dataset with other high-quality instruction data might yield more promising results for these larger 70B models. Due to constraints in computational resources and scheduling, our studies on the data mixing strategy for these advanced 70B models are insufficient. We plan to address this limitation in future work.

## B Differences between Recent Works

This section will discuss the primary differences between our designed data generation pipeline and existing works, like Prometheus (Kim et al., 2024).

**Difference in Data Preparation**   Although Prometheus collects five responses with quality scores ranging from 1 to 5 (Kim et al., 2024), these responses are synthesized using a GPT-4 reference response, leading to responses that are very similar to the reference, which is a significant deviation from real-world scenarios.

**Difference in Crucial Information**   Although Prometheus also employs customized criteria for better critiques (Kim et al., 2024), our work differs significantly. Our evaluation criteria are organized into a hierarchical two-tier structure, providing clear definitions for diverse evaluation dimensions—a method proven effective in automatic evaluation (Lee et al., 2024; Liu et al., 2024e). In contrast, Prometheus synthesizes one criterion using GPT-4, lacking sufficient guidelines for high-quality reference response and critique generation.

## C Implementation Details

### C.1 MultiCritiqueDataset Construction

**Query Preparation**   All the queries in Auto-J (Li et al., 2024b) and DEITA (Liu et al., 2024b) are collected. For OpenHermes-2.5[4], we sample 1K queries for its 28 categories, leading to 28K queries. Following previous work (Yuan et al., 2024), we use 3.2K examples from the OpenAssistant dataset by sampling only the first conversation turns in the English language that achieves the highest human-annotated scores. Besides, we also sample 2K mathematical and coding questions from MetaMathQA (Yu et al., 2024) and CodeFeedback (Zheng et al., 2024) datasets to collect critiques for reasoning tasks. Only the first conversation utterance (the coding question) in CodeFeedback is used. **None of the training samples are from the test set in CriticEval and CriticBench benchmarks.**

**Collect Evaluated Responses**   To collect diverse evaluated responses for queries, we use eleven widely-used LLMs with varying scales and capabilities in this work: (1) Qwen-1.5-72B-Chat; (2) Qwen-1.5-7B-Chat; (3) InternLM2-20B-Chat; (4) Yi-34B; (5) Mixtral-8x7B-Instruct; (6) Llama2-13B-Chat; (7) Llama2-7B-Chat; (8) Gemma-2B; (9) Baichuan2-13B-Chat; (10) Vicuna; (11) WizardLM-7B-v0.1. The LMDeploy tookit (Contributors, 2023a) is used to inference these LLMs by random sampling decoding method, and the hyper-parameters are 0.95 top-p and 0.8 temperature. Besides, the InternLM2-20B-reward (Cai et al., 2024) model[5] is used to score the quality of

---

[4] https://huggingface.co/datasets/teknium/OpenHermes-2.5

[5] InternLM2-20B-Reward was the top-tier reward model in RewardBench (Lambert et al., 2024) when we start our project.

responses, and the reward scores are used to classify responses into three quality levels. Our preliminary experiments reveal that this reward model exhibits a strong correlation with human judgments in distinguishing response quality. For example, in discriminating between 200 pairs of high/low quality responses, InternLM2-20B-reward achieved a 95.3% consistency with human judgments. Therefore, we use the reward model to automatically complete this process. The average reward scores for each response quality are shown in Table 9. It can be observed that there exists a significant performance gap among these response qualities.

Given that reward models fail to accurately evaluate the quality of responses in mathematical and coding questions, we only collect two kinds of response qualities: (1) high-quality responses generated by GPT-4o and (2) low-quality responses generated by eight 7B-20B open-source LLMs.

| Resp. Quality | Avg. |
|---|---|
| **Low** | -1.41 |
| **Medium** | 0.70 |
| **High** | 1.69 |

Table 9: The average reward model scores for each response quality.

**Collect Crucial Information**    The prompt for LLMs to generate task description, two-tier structured criteria and reference response are described in Appendix J. Our preliminary study reveals that reference responses tend to produce critiques that lack diversity for mathematical and coding questions. As a result, we set reference responses as empty for these two tasks.

Most previous works rely on human-annotated criteria for each task (Hu et al., 2024; Li et al., 2024b), which do not scale well. We propose generating a customized two-tier structure evaluation criteria tailored to each query using GPT-4. Besides, the user pre-defined criteria are provided as input optionally for better flexibility.

**Pre-process Evaluated Responses**    Our proposed ACUs contain the location of flaws in the evaluated response for better interpretability. To achieve this goal, we pre-process the evaluated responses by appending labels for sentences in evaluated responses. For most tasks, punctuation marks such as periods, exclamation marks, and semicolons are used to divide sentences. For code-related task scenarios, the sentence is divided by the line breaks to represent lines of the evaluated code.

**Collect Preference Dataset**    The threshold of differences in accumulated severity scores is set as 5 in this paper. Besides, we leverage four additional 7B LLMs to revise the evaluated response eight times, given the model-generated critiques: (1) InternLM2.5-7B-Chat; (2) Llama-3.1-8B-Instruct; (3) Qwen2-7B-Chat; (4) Mistral-7B-Instruct. The random sampling decoding method is used to generate diverse revisions, and the hyper-parameters are (1) 0.95 top-p, (2) 50 top-k, and (3) 1.0 temperature. The vLLM toolkit (Kwon et al., 2023) is used to speed up the inference.

### C.2    EXPERIMENTAL DETAILS

**SFT**    During the SFT training stage, the InternLM2-7B-Chat model is fine-tuned by optimizing the Maximum Likelihood Estimation (MLE) loss:

$$L_{\text{MLE}} = -\frac{1}{\mathbb{N}} \sum_{i=1}^{\mathbb{N}} \log p_\theta(\mathcal{CI}_i, \mathcal{C}_i | q_i, r_i) \qquad (2)$$

The training process is running on 2 A800 GPU Serves (16 GPUs) by using DeepSpeed[6]. To achieve a fair comparison, we set the training hyper-parameters as follows: (1) 4e-5 learning rate; (2) 6e-6 minimum learning rate; (3) 32,768 maximum sequence length; (4) 2 epoch; (5) 1 batch size; (6) AdamW optimizer.

We explore the effect of instruction format and data recipe of crucial information during the SFT stage in Appendix I. We fix the following experimental setup for supervised fine-tuning: (1) the proportion of the single-turn template is 5% and left 95% training samples for SFT are multi-turn conversations, consisting task description, two-tier structured evaluation criteria, reference responses, critiques consisting of a list of ACUs generated by MultiCritique-SFT pipeline and summarization of the final judgment for the evaluated response; (2) the crucial information for each query is only optimized once in 2 epochs.

---

[6]https://github.com/microsoft/DeepSpeed

In Section 6, we analyze the contributions of our proposed MultiCritique-SFT pipeline. We only collect the summarization of final judgments for the critiques generated by MultiCritique-SFT , and the critiques generated by each LLM do not have the corresponding summarizations. Thus, in this experiment, we do not fine-tune the model to predict the summarization of final judgments in 95% multi-turn training samples.

**Ablation Study in SFT** We conduct the ablation study in Section 6 (**Ablation Study on MultiCritique-SFT**) to prove the effectiveness of aggregated critiques generated by our proposed MultiCritique pipeline. The dataset in this ablation study is slightly different from that in the main experiment, consisting of two parts:

- **Distinct parts**: crucial information and analytical critiques (a list of ACUs), without the summarization and judgment score.
    - **MultiCritique-SFT Critiques:** Critiques are generated through our MultiCritique-SFT pipeline, which aggregates accurate ACUs from multiple models via meta-critique classification.
    - **Four Individual LLMs Critiques:** Critiques are extracted from our raw MultiCritique-SFT dataset, using feedback generated independently by four models.
- **Shared parts**: to enable the objective evaluation on CriticEval and CriticBench benchmarks, we supplement 5% of samples from MultiCritique-SFT to ensure the fair comparison, which consists of crucial information, analytical critiques, summarization, and judgment score.

**Reinforcement Learning** During the reinforcement learning stage, we first train the InternLM2-7B-Chat as the reward model on MultiCritiqueDataset-RL by using xtuner toolkit (Contributors, 2023b), and the hyper-parameters are as follow: (1) 32,768 maximum sequence length; (2) 1 epoch; (3) 1 batch-size; (4) AdamW optimizer; (5) 2e-5 learning rate; (6) focal loss (Lin et al., 2018). For the $i$-th sample, the focal ranking loss is computed to optimize the reward model:

$$L_{\text{ranking}} = -(1 - 2 \times \max(0, P^i_{j_+,j_-} - \frac{1}{2}))^2 \log(P^i_{j_+,j_-}), \quad (3)$$

where $P^i_{j_+,j_-} = \sigma(r^i_{j_+} - r^i_{j_-})$ represents the probability that the reward score of $C_{i,j_+}$ is greater than that of $C_{i,j_-}$. The difficulty decay coefficient only takes effect when the model correctly predicts the preference of $i$-th training sample, *i.e.,* $P^i_{j_+,j_-} > 0.5$, otherwise it equals to 1.

Subsequently, we conduct the PPO algorithm to optimize the SFT model on six nodes of A800 GPU servers (48 GPU cards) with the ray toolkit.[7] The hyper-parameters during reinforcement learning are listed as below: (1) 30,000 maximum sequence length; (2) 64 batch-size; (3) deepspeed zero-2; (4) 0.9 top-p and 1.0 temperature sampling parameters for policy model.

**Evaluation** We leverage the publicly available codebase of CRITICEVAL and CRITICBENCH for evaluation. To ensure the robust objective evaluation of the revision critique dimension, we leverage five LLMs with varying capabilities to revise the responses given feedback generated by each baseline: InternLM2-7B-Chat, InternLM2.5-7B-Chat, InternLM2-20B-Chat (Cai et al., 2024), Mixtral-7x8B-Instruct (Jiang et al., 2024) and Llama-3.1-70B-Instruct. Due to the limited OpenAI API budget, we only conduct the subjective evaluation on the revision dimension to evaluate the quality of revisions generated by the Llama-3.1-70B-Instruct model.

In CRITICBENCH benchmark, the responses with $\geq 7$ Likert Scores generated by our fine-tuned models are treated as the positive samples since responses with $\geq 7$ are comparable or better than the reference answers in our defined score rubrics, which is described in Appendix J.6. The responses with $> 2$ quality scores are treated as positive samples for the Prometheus model since the overall score range is 1 to 5.

Regarding the general capability evaluation in Section 6, we evaluate 18 objective evaluation benchmarks. The complete results are shown in Table 10. Experimental results that LLM's performance on these objective benchmarks is slightly improved with our proposed MultiCritiqueDataset-SFT.

---

[7]https://github.com/ray-project/ray

| Benchmark | w/o MultiCritique Dataset-SFT | w/ MultiCritique Dataset-SFT |
|---|---|---|
| MMLU (Hendrycks et al., 2021a) | 62.28 | **62.57** |
| CMMLU (Li et al., 2024a) | 61.13 | **61.22** |
| C-Eval (Huang et al., 2023) | 57.91 | **58.34** |
| GaoKaoBench (Zhang et al., 2024d) | **55.89** | 55.13 |
| TriviaQa (Joshi et al., 2017) | 67.56 | **67.66** |
| NQ (Kwiatkowski et al., 2019) | **27.04** | 26.23 |
| RACE (Lai et al., 2017) | **88.16** | 88.08 |
| Winogrande (Sakaguchi et al., 2019) | **74.59** | 73.32 |
| HellaSwag (Zellers et al., 2019) | **93.38** | 93.36 |
| BBH (Suzgun et al., 2022) | **60.92** | 60.3 |
| GSM8K (Cobbe et al., 2021) | **75.44** | 74.3 |
| MATH (Amini et al., 2019) | 41.92 | **42.72** |
| TheoremQA (Chen et al., 2023) | 15.75 | **16.88** |
| HumanEval (Chen et al., 2021) | 56.1 | **57.93** |
| MBPP (Austin et al., 2021) | 55.25 | **57.59** |
| CodeBench (LCBench) | **16.07** | 12.95 |
| GPQA (Rein et al., 2023) | 26.77 | **29.8** |
| IFEval (Zhou et al., 2023) | 58.04 | **59.33** |
| Average | 55.23 | **55.43** |

Table 10: The complete list of objective general benchmarks.

To ensure reproducibility, the greedy search decoding strategy is used for inference. As for the models we fine-tuned on our proposed MultiCritiqueDataset, the optional user pre-defined criteria are empty during inference.

# D STATISTICS OF MULTICRITIQUEDATASET

The statistical information of our proposed MultiCritiqueDataset is shown in Table 11. Our proposed MultiCritiqueDataset significantly outperforms existing critique datasets from multiple dimensions, like response quality and the number of tasks. Although the size of UltraFeedback and Feedback-Collection are greater than our proposed MultiCritiqueDataset, the models fine-tuned on them are much worse than that fine-tuned on MultiCritiqueDataset, demonstrating the better quality of our proposed dataset. Although Feedback-Collection and Preference-Collection consist of 5 response qualities, they are synthesized by GPT-4, resulting in very similar content with reference responses.

| Dataset | Type | Task Desc. | Criteria | Ref. | Tokens | Resp. Quality | Num. Task | Num. Query | Num. Resp. | Avg. Turn | Public |
|---|---|---|---|---|---|---|---|---|---|---|---|
| Auto-J | SFT | ✗ | ✗ | ✗ | 3.8M | - | 58 | 4.4K | 4.4K | 1 | ✓ |
| UltraFeedback | SFT | ✗ | ✗ | ✗ | 227M | - | 9 | 257K | 257K | 1 | ✓ |
| TIGERScore | SFT | ✗ | ✗ | ✗ | 23.7M | - | - | 42.5K | 42.5K | 1 | ✓ |
| Feedback-Collection | SFT | ✗ | ✓ | ✓ | 191.5M | 5 | - | 20K | 100K | 1 | ✓ |
| Preference-Collection | SFT | ✗ | ✓ | ✓ | 382.9M | 5 | - | 40K | 200K | 1 | ✓ |
| Themis | SFT,RL | ✗ | ✓ | ✗ | - | - | 9 | 67K | 67K | 1 | ✗ |
| JudgeLM | SFT | ✗ | ✗ | ✓ | - | - | - | 100K | 200K | 1 | ✓ |
| MultiCritiqueDataset-SFT | SFT | ✓ | ✓ | ✓ | 531.1M | 3 | 123 | 10.7K | 32.1K | 2.40 | ✓ |
| MultiCritiqueDataset-RL | RL | ✓ | ✓ | ✓ | 352.9M | 3 | 123 | 19.7K | 39.4K | 2.35 | ✓ |

Table 11: The comparison between our proposed MultiCritiqueDataset and existing critique datasets. **Avg. Turn** represents the average number of utterances in the multi-turn conversation history, and the user query is the last utterance in it. The number of tokens is counted based on the InternLM2-7B-Chat tokenizer.

The complete list of the task scenarios in our proposed MultiCritiqueDataset is shown in Table 12, consisting of 123 tasks. Except for 58 fine-grained tasks defined in Auto-J (Li et al., 2024b), our proposed dataset includes 65 categories defined in the OpenHermes-2.5 dataset.

| Task | Num. | Task | Num. | Task | Num. |
|---|---|---|---|---|---|
| default | 9362 | math reasoning | 6228 | code generation | 5280 |
| explaining general | 3452 | open question | 3048 | seeking advice | 2674 |
| value judgement | 2586 | roleplay | 1210 | functional writing | 958 |
| verifying fact | 838 | brainstorming | 828 | analyzing general | 780 |
| code correction rewriting | 720 | chemistry | 718 | physical | 710 |
| chitchat | 702 | bio | 702 | asking how to question | 690 |
| creative writing | 632 | rejecting | 540 | planning | 538 |
| counterfactual | 528 | awareness | 480 | editor | 480 |
| misconception | 480 | general | 480 | cot | 480 |
| experience | 480 | song | 480 | plan | 480 |
| joke | 480 | rp | 480 | multiple choice | 480 |
| trivia | 480 | counterfactual contextual | 478 | stylized response | 478 |
| theory of mind | 478 | writing | 478 | greeting | 478 |
| orca | 478 | riddle | 478 | wordgame | 478 |
| gtkm | 468 | recommendation | 462 | solving exam question without math | 456 |
| coding | 452 | writing personal essay | 432 | text summarization | 430 |
| summarization | 424 | explaining code | 408 | agent | 406 |
| text to text translation | 396 | writing email | 372 | question generation | 372 |
| card | 372 | instructional rewriting | 360 | ranking | 358 |
| writing song lyrics | 318 | writing cooking recipe | 314 | information extraction | 300 |
| post summarization | 300 | data analysis | 294 | writing job application | 294 |
| writing presentation script | 292 | classification identification | 276 | solving exam question with math | 276 |
| paraphrasing | 240 | detailed writing | 222 | writing advertisement | 142 |
| writing social media post | 138 | title generation | 132 | text correction | 120 |
| language polishing | 114 | writing product description | 108 | writing blog post | 96 |
| code to code translation | 92 | writing legal document | 90 | writing technical document | 74 |
| reading comprehension | 66 | text simplification | 60 | writing scientific paper | 48 |
| keywords extraction | 40 | writing marketing materials | 36 | topic modeling | 18 |
| writing news article | 18 | quiz | 18 | writing chapter | 16 |
| code simplification | 12 | note summarization | 12 | writing letter | 12 |
| writing history essay | 6 | predicting general | 6 | writing feature story | 6 |
| criticism | 6 | challenges | 6 | writing social responsibility report | 6 |
| impact | 6 | impact analysis | 6 | changing mindset | 6 |
| overview | 6 | writing consumer complaint | 6 | writing dialogue | 6 |
| writing sequel | 6 | writing historical document | 6 | exit planning | 6 |
| writing screenplay | 6 | writing deployment script | 6 | data conversion | 6 |
| time zone conversion | 6 | language history | 6 | writing press release | 6 |
| writing survival manual | 6 | writing movie review | 6 | writing biography | 6 |
| reward | 6 | writing comedy skit | 6 | writing note | 6 |
| writing love note | 6 | writing love letter | 6 | writing config file | 6 |
| writing script | 6 | writing kubernetes deployment file | 6 | writing code | 2 |

Table 12: The complete list of task scenarios in our proposed MultiCritiqueDataset-SFT. The number of samples is also listed.

The overall quota for using OpenAI and Claude API to construct our proposed MultiCritiqueDataset are 9,180\$ and 125.6\$, respectively. The average API cost for each sample is 0.29\$. Given that the average price of one human-annotated critique is 8\$ (Wang et al., 2023), our data generation pipeline is much cheaper and easier to scale to more diverse task scenarios.

# E EXCLUDED BASELINES AND BENCHMARKS

## E.1 EXCLUDED BASELINES

Some baselines are excluded during our evaluation, and the reasons are described as follows: (1) InstructScore (Xu et al., 2023) is trained on samples with limited tasks, failing to extend to other diverse tasks, like mathematics reasoning and code generations; (2) JudgeLM (Zhu et al., 2023) is mainly trained to compare two responses with critiques. Although it can be used to score the single responses, the reference responses should be supplied[8], which are unavailable in CRITICEVAL and CRITICBENCH; (3) Reward models (Lambert et al., 2024) are also widely used to evaluate the quality of responses. However, their scores can only reflect the relative differences in response quality, so reward models cannot be assessed in CRITICBENCH. Additionally, due to the lack of textual critiques, reward models are unsuitable for evaluation under CRITICEVAL.

## E.2 EXCLUDED BENCHMARKS

Existing benchmarks for evaluating the critique ability of LLMs could be classified into two categories: (1) single-response evaluation; (2) pairwise response comparison (Li et al., 2024b; Kim et al., 2024). Single-response evaluation aims to evaluate the quality of a single response given the context of the conversation or user query. For example, CRITICEVAL and CRITICBENCH evaluate whether LLMs could accurately score the quality of responses. Pairwise response comparison selects the better response from a pair of responses. For example, RewardBench (Lambert et al., 2024), Feedback-Bench (Kim et al., 2024) and PandaLM test set (Wang et al., 2024c) consist of numerous pairs of responses with clear performance gap. Since pairwise response comparison is much simpler than comparing the scores corresponding to the two responses, it is unfair for our models under the pairwise response comparison benchmarks. Therefore, these benchmarks are not used in our paper. We will extend our proposed MultiCritiqueDataset from single-response evaluation to the pairwise response comparison (Li et al., 2024b).

# F PRELIMINARY STUDY ON CRUCIAL INFORMATION

During designing our data generation pipeline, we conducted a preliminary study to verify whether crucial information helps reduce the complexity of critique tasks and improve the quality of collected critiques. Specifically, we conducted the self-critique prompting (Pan et al., 2024) to collect critiques and corresponding revisions and evaluated the quality of the critiques by measuring the quality of their corresponding revisions.

|  | Reward |
|---|---|
| **Ref. w/ Criteria** | 0.54 |
| **Ref. w/o Criteria** | 0.45 |

Table 13: Avg. rewards.

## F.1 EXPERIMENTAL SETUP

We first random sample 1,280 queries and evaluated responses from MultiCritiqueDataset. Then, four LLMs are prompted to generate critiques and subsequent revisions with or without each crucial information: (1) InternLM2.5-7B-Chat; (2) Qwen2-7B-Chat; (3) Llama-3.1-8B-Instruct; and (4) Mixtral-7B-Instruct. Each model generates critiques and revisions eight times, leading to overall $4 \times 8 = 32$ revisions, and the advanced InternLM2-20B-reward model judges the quality of these revisions.

|  | Reward |
|---|---|
| **Origin Response** | -0.11 |
| **w/ All** | **0.085** |
| **w/o Task** | 0.076 |
| **w/o Ref.** | 0.029 |
| **w/o All** | -0.005 |

Table 14: Avg. reward scores.

## F.2 EXPERIMENTAL RESULTS

First of all, as shown in Table 13, it can be found that the quality of reference responses generated given the customized evaluation criteria is much better, indicating the effectiveness of our proposed two-tier structure evaluation criteria.

---

[8]https://github.com/baaivision/JudgeLM/tree/main/judgelm/llm_judge

Besides, as shown in Table 14, it can be found that the quality of revisions becomes worse when task descriptions and reference answers are removed. Besides, removing all the crucial information leads to the worst performance (-0.005 < 0.085). Note that we do not evaluate the contributions of customized evaluation criteria since its contribution is proven in Table 13, *i.e.,* improves the quality of reference responses.

## G    CAN OTHER LLMS CONDUCT META-CRITIQUE?

Our MultiCritique framework is a general and model-agnostic framework. GPT-4 serves as one possible meta-critique judge model. While we chose GPT-4 due to its advanced meta-critique capabilities (Sun et al., 2024; Lan et al., 2024), any sufficiently advanced LLM can fulfill this role. To demonstrate this flexibility, we conducted additional experiments with Claude-3.5-Sonnet and Qwen2.5-72B-Instruct.

| Model | Corr. | Agree. |
|---|---|---|
| **Claude-3.5-Sonnet** | 0.58 | 0.71 |
| **Qwen2.5-72B-Instruct** | 0.60 | 0.72 |

Table 15: Correlation and agreement between advanced LLMs and GPT-4.

Specifically, we randomly sample 200 samples from MultiCritiqueDataset-SFT, and prompt Qwen2.5-72B-Instruct and Claude-3.5-Sonnet to re-run the Meta-Critique Classification and Critique Summarization processes. Subsequently, we assess two metrics to reflect the correlations between GPT-4 and these models: (1) Spearman correlation scores between GPT-4 and these models on assessing the response quality; (2) Agreement on judging the correctness of each ACU in Meta-Critique classification. As shown in Table 15, these models achieve not only high meta-critique agreement ($> 70\%$, the random baseline is 50%) with GPT-4 but also strong correlation (Spearman $\rho \approx 0.60$) in final judgment scores. These results indicate that the effectiveness of our MultiCritique does not critically depend on GPT-4. Other advanced models could also be used in MutiCritique to conduct the Meta-Critique classification.

## H    EXPERIMENTS ON MORE LLMS

Except for InternLM2-7B-Chat-SFT, we also conduct experiments on three more advanced LLMs: (1) Llama-3-8B-Instruct (Meta, 2024); (2) Qwen2.5-7B-Instruct (Team, 2024); and (3) InternLM2.5-7B-Chat (Cai et al., 2024). Table 16 demonstrates that MultiCritique-SFT also effectively improves the critique ability of these advanced 7B LLMs. Notably, the fine-tuned Qwen2.5-7B-Instruct model outperforms GPT-4 on CRITICBENCH (79.04% > 78.75%).

| Models | CRITICEVAL | CRITICBENCH | | | | | |
|---|---|---|---|---|---|---|---|
| | $F_{obj.}$ | Math | Comm. | Symb. | Algo. | Code | Overall |
| Qwen2.5-7B-Instruct | 62.50 | 87.54 | 55.60 | 65.41 | 50.96 | **84.05** | 74.22 |
| + MultiCritiqueDataset-SFT | **64.82** | **93.45** | **60.59** | **66.67** | **62.96** | 82.06 | **79.04** |
| Llama3-8B-Instruct | 37.20 | 78.33 | **62.64** | **62.05** | 62.19 | 76.41 | 70.71 |
| + MultiCritiqueDataset-SFT | **51.87** | **87.00** | 59.92 | 61.41 | 60.22 | 74.61 | **73.92** |
| InternLM2.5-7B-Chat | 44.84 | 59.46 | **63.97** | 48.26 | 42.63 | 37.21 | 53.98 |
| + MultiCritiqueDataset-SFT | **58.29** | **88.56** | 62.13 | **57.02** | **57.35** | 78.37 | **73.72** |

Table 16: Evaluation of critique ability of advanced Qwen2.5-7B-Instruct, Llama-3-8B-Instruct and InternLM2.5-7B-Chat models.

## I    HOW FACTORS AFFECT PERFORMANCE DURING SFT STAGE

In this section, we analyze two factors that influence the performance of models fine-tuned on our proposed MultiCritiqueDataset-SFT: (1) instruction format; and (2) the data recipe of crucial information.

### I.1    INSTRUCTION FORMAT

Our proposed MultiCritiqueDataset-SFT consists of mult-turn conversations for generating critiques. To ensure the generalization of fine-tuned models, in this paper, we construct the single-turn and multi-turn prompt templates in the instruction dataset for the supervised fine-tuning (SFT)

stage. Our experimental results reveal that the proportion of single-turn and multi-turn templates in the training data significantly affects the model's performance. As shown in Table 17, it can be observed that when the proportion of single-turn templates is 5% of the total data size, the fine-tuned models could achieve optimal performance on the feedback objective evaluation of CRITICEVAL during the SFT stage. Therefore, this setting is used in all the experiments in our paper.

## I.2 DATA RECIPE OF CRUCIAL INFORMATION DURING SFT STAGE

As described in Appendix D, our proposed MultiCritiqueDataset-SFT consists of 32.1K evaluated responses and 10.7K queries, and the same query has the same crucial information: task description, criteria, and reference response. Therefore, the training volume on crucial information will be three times larger than that of critiques. This might lead to overfitting crucial information, influencing the optimization of critiques. To address this problem, we mask the loss of crucial information at varying rates. As shown in Table 18, it can be found that the performance of fine-tuned model on CRITICEVAL benchmark decreases when the proportion of training volume on crucial information increases, and the best proportion of training volume is 16.67%, *i.e.,* **the crucial information for each query is only optimized once in 2 epochs**. We leverage this experimental setting in all our experiments.

| Rate | $F_{obj.}$ |
|---|---|
| **1.0%** | 61.14 |
| **2.5%** | 57.32 |
| **5.0%** | **63.85** |
| **10.0%** | 60.21 |

Table 17: The proportion of single turn prompts.

| Rate | $F_{obj.}$ |
|---|---|
| **16.67%** | 63.85 |
| **33.33%** | 60.19 |
| **66.67%** | 56.99 |
| **100.0%** | 57.47 |

Table 18: The proportion of training volume on crucial information.

## J DESIGNED PROMPTS IN MULTICRITIQUE

In this section, we provide the detailed prompts that used in our proposed MultiCritiqueDataset data generation pipeline.

### J.1 TASK DESCRIPTION

The prompt for GPT-4 model to generate the task description is shown in Figure 3, while the multi-turn conversations are not provided.

---

Now, you are a helpful assistant aiming to provide valuable critiques and analysis for the previous conversation history, thereby assisting in the analysis of the quality of subsequent responses in relation to this conversation history history.

# Your Tasks
Analyze and describe the primary purpose of user's query in conversation history. Do NOT generate very lengthy description, keep it concise and precise. **If the conversation history contains multiple turns between assistant and human, MUST analyze the main purpose of the user's last query by considering the previous conversation history.**

# Output Template
Generate the task description in following Markdown template. Do NOT add comment (//) in the template.
—
// a string for task description
# Task Description
A string analyze the attribute of the task
—

---

Figure 3: The prompt for generating task description about the last user query in conversation.

### J.2 CRITERIA GENERATION

The prompt for GPT-4 to generate the two-tier structured criteria is shown in Figure 7. Note that the user could provide their pre-defined criteria. If the criteria provided by users are not empty, GPT-4 is asked to generate the two-tier structured criteria from scratch; otherwise, GPT-4 is asked

to expand on the criteria provided by the user and must not generate content that conflicts with the user's provided criteria. Besides, it can be found that each item of criteria consists of 3 fundamental values: (1) criteria name, (2) criteria fine-grained description, and (3) importance degree of the criteria (normal, medium, important).

## J.3 REFERENCE GENERATION

Given the two-tier structured criteria, GPT-4 is asked to generate high-quality reference responses that satisfy all the evaluation criteria, as shown in Figure 4.

> # Task Goal
> Good! Your task is to generate a high-quality response for the **conversation history (before we provided the criteria list)**, which perfectly satisfies all the generated **first-tier and second-tier** criteria in last turn.
>
> # NOTICE!!!
> **1. The conversation history here represents the conversations before we provided the criteria list. Do NOT respond to the last utterance.**
> **2. Do NOT generate any explanation or analysis about your generated response.**

Figure 4: The prompt for generating reference response given the criteria.

## J.4 MULTI-AGENT ANALYTICAL CRITIQUE

After generating the three crucial information, multiple LLMs are asked to follow the instructions in Figure 8 to critique the evaluated responses. It can be found that LLMs are asked to critique the evaluated responses sentence by sentence and generate a list of Analytical Critique Units (ACUs) consisting of 5 key values: (1) citation symbol of the sentence in evaluated response; (2) description of this flaw; (3) which criteria this flaw belongs to; (4) severity of this flaw; (5) revision suggestions.

## J.5 META-CRITIQUE CLASSIFICATION

As shown in Figure 5, after collecting multiple critiques generated by LLMs, the GPT-4 model is asked to conduct the meta-critique to analyze the quality of each ACU. Each ACU is classified into seven categories.

The detailed descriptions of each meta-critique and corresponding severity score are shown in Table 19.

| Label | Meaning | Detailed Description of Quality Category | Severity (1-5) |
|---|---|---|---|
| L0 | Correct ACU | This feedback is accurate and provide helpful suggestions. | 0 |
| L1 | False Negative ACU | The content is free from any flaws and should not be critiqued. | 5 |
| L2 | Wrong Criteria | The type of criteria of feedback is misclassified or does not exist. | 2 |
| L3 | Wrong Severity | The severity of this flaw is misclassified. | 1 |
| L4 | Wrong Description | The descriptions of flaws are unreasonable or inaccurate. | 4 |
| L5 | Wrong Suggestion | The suggestions for revisions are unreasonable or introduce errors. | 4 |
| L6 | Unhelpful Suggestion | Revision suggestions are reasonable but not helpful. | 1 |

Table 19: Our human-annotated quality categories of ACUs. A higher severity score indicates the worse performance of corresponding ACUs.

## J.6 CRITIQUE SUMMARIZATION

Finally, the GPT-4 model is asked to summarize the critiques from multiple LLMs and generate the final critiques and summarization for the evaluated responses. As shown in Figure 6, it can be found that the judgment scores for evaluated responses are the floating numbers ranging from 1 to

Good! Now, I want you to carefully re-check (meta-evaluation) each feedback entry generated by these models.

## Categories of Errors in Feedback Entries
Please carefully analyze each feedback entry in this list sequentially and categorize them into the following error types based on their errors:
E0. the feedback entry is helpful, perfect, and satisfying and accurately points out the flaw in the response, providing helpful suggestions for improvement.
E1. the cited sentence in the feedback entry is good without any flaws belonging to the mentioned criteria, and it should not be critiqued for the mentioned criteria.
E2. the cited sentence in the feedback entry has flaws belonging to the mentioned criteria, but the type of criteria is misclassified or does not exist in the previous criteria list.
E3. the severity of this flaw is misclassified.
E4. the description of this flaw is unreasonable and inaccurate.
E5. the suggestions for revising this flaw are unreasonable or introduce new problems.
E6. although revision suggestions for the flaw are reasonable without any problems, revision with suggestions will not necessarily improve the quality of the response.

## NOTICE!!!
1. Ensure the number of the generated analysis entries equals the number of feedback entries generated by the corresponding model. **Do NOT miss any feedback entries for analysis.**
2. If one feedback entry is similar to or the same as some analyzed feedback entries, **Do NOT regard it as a redundant feedback entry (redundant error). Please evaluate this feedback entry by focusing on analyzing errors (E0 to E6) in the feedback entry content.**

Please analyze each feedback entry one by one and sequentially, which will be used to summarize the final feedback generation.

Figure 5: The prompt for generating meta-critiques for all the critiques generated by multiple LLMs.

10, and the $\geq 7$ scores indicate the comparable and even better qualities of responses than reference responses.

# Task Goal
Your goal is to summarize your final feedback entry list based on your meta-evaluation decisions. In your meta-evaluation decision, you have carefully analyzed all the feedback generated by various models and decided which feedback entries should be included in your final feedback entry list in the last conversation turn.

# Your Task
## 1. Reorganize the Helpful Feedback Entry List
**Now, please reorganize the previous output and strictly abide by the following notes**:
(1) Include all the feedback entries from all the models you think are helpful and have been considered "Yes" for inclusion. **Do NOT miss any helpful and essential feedback entries**;
(2) Appropriately summarize and consolidate multiple feedback entries with the same cited sentences from different models into one feedback entry. Ensure the summarized descriptions and suggestions contain helpful details in these multiple feedback entries. Also, ensure that the final feedback entry list does not have numerous feedback entries with duplicate content;
(3) If a flaw is labeled as E6 (not helpful for improvement) and the meta-evaluation acknowledges it, it is optional whether to remove this feedback entry based on your preference. Always remember your goal is to generate "helpful and valuable" feedback entries that are beneficial for refinement;
(4) If some problematic feedback entries (not labeled as E0 or the consideration is "No") could become more reasonable and valid after being revised according to your meta-evaluation description, and these feedback entries have not been considered in other helpful feedback entries, please also revise these feedback entries and supplement them to your final output;
(5) Each feedback entry contains only one criteria. Do NOT assign multiple criteria to one feedback entry. If the sentence has numerous flaws, please list them in multiple feedback entries.

## 2. Summarize
### 2.1 Summarize Your Analysis
Please summarize and describe the performance of evaluated response on each first-tier primary criteria.
### 2.2 Generate Your Judgements
In the end, you should provide your final judgement score, ranging from 1 to 10. The score ranges and definitions are shown as follows:
1. $1 \leq x < 3$: The quality is very low, containing numerous severe flaws; there are also other flaws, with Important error criteria.
2. $3 \leq x < 5$: The quality is low, making it difficult to fulfill user query; There are many flaws, and a small number of severe flaws may be included.
3. $5 \leq x < 7$: The quality is moderate, somewhat addressing the user query; There are a few errors, and a small number of severe errors may be included.
4. $7 \leq x < 9$: The quality is approximately the same as the reference response (with the reference response scoring around 8). The response effectively answers user query.
5. $9 \leq x < 10$: The quality is better than the reference, perfectly answering the user query in the conversation history.

## NOTICE!!!
1. Quality scores (1-10) can be expressed as floating-point numbers.
2. Within specific score ranges, the more flaws there are, the lower quality score, and vice versa.
3. You should compare the evaluated response the reference before giving your quality score. Please follow the important guideline as follows: if evaluated response is worse than the reference, its score should be lower.

Figure 6: The prompt for generating final critiques and summarization for the evaluated responses, which is used for the supervised fine-tuning stage.

# K  CASE STUDY IN MULTICRITIQUEDATASET

## K.1  CASE STUDY OF CUSTOMIZED EVALUATION CRITERIA

We provide one case of two-tier structured evaluation criteria for one query in MultiCritiqueDataset in Figure 9. Compared with existing works, like Themis and Auto-J, our evaluation criteria contain a more diverse and customized evaluation dimension for the user query, which is beneficial for robust and accurate evaluation.

## K.2   CASE STUDY OF CRITIQUES

We provide one case of analytical critique units (ACU), summarization, and judgment of critiques in Figure 10. Each ACU points out one flaw in a located sentence in the evaluated responses.

Now, you are a helpful assistant aiming to provide valuable critiques and analysis for the previous conversation history, thereby assisting in the analysis of the quality of subsequent responses in relation to this conversation history history. Now, we have provided our criteria list (maybe empty) for you from different evaluation perspectives as below.
—

# Our Provided Criteria List
{user_pre_defined_criteria}
—

# Your Tasks
## Supplement and Decompose the Criteria
Generate the criteria list of the two-tier structure: (1) The first-tier structure consists of primary criteria, i.e., the evaluation dimensions broadly conceptualized and distinct based on conversation history; (2) The second-tier structure decomposes these primary evaluation dimensions into several fine-grained and precise criteria based on the information in conversation history. **Note that our provided criteria list are only the primary criteria list (first-tier) without the fine-grained criteria definition (second-tier).**

### 2.1 If our provided criteria list is **EMPTY**

Please directly generate this two-tier criteria structure from scratch.
**Do NOT generate redundant criteria; keep the final criteria precise, helpful, and concise.**

### 2.2 If our provided criteria list is **NOT EMPTY**

**Firstly, you should keep all our provided criteria as the primary criteria in your final output.** You could expand other primary criteria not considered in our provided criteria but are essential for analyzing flaws in responses for previous conversation history.
1. **But NEVER expand primary criteria that conflict with our provided criteria.**
2. **NEVER generate criteria that are redundant with our provided criteria.**
3. **Do NOT miss any criteria that exists in our provided criteria list.**
Secondly, you should decompose these primary criteria into several fine-grained and precise criteria by considering the conversation history.

### 2.3 NOTICE!!!
**Keep the number of all fine-grained criteria within 15, and each primary criterion includes no more than 3 fine-grained criteria.**

# Output Template
Generate the task description in following Markdown template. Do NOT add comment (//) in the template.
—
# Two-tier Structure of Criteria
// a block for one primary criteria consisting of no more than 3 fine-grained criteria. Keep output following structure in order. Variable in '{{}}' should be replaced.
## {{Name of First Primary Criteria}}
// a string of the description and details of this first-tier primary criteria
Description: {{description}}

### {{Name of Fine-grained Criteria}}
// a string of the description and details of this second-tier
fine-grained criteria
Description: {{description}}
// a word reflects the significance of fine-grained criteria, select degree from three types (least to most significance): (1) normal; (2) medium; (3) important Degree: {{degree}}
...
—

Figure 7: The prompt for generating two-tier structured criteria. We also allow user to input their specific evaluation criteria.

# Task Input
We provide the evaluated response that responds to the conversation history as below.
—
{evaluated_response}
—

## NOTICE!!!
**1. The conversation history represents the conversations before we provided the criteria list.**
**2. The evaluated response contains citation symbols, like [S1] and [S2] ([S1] means sentence 1), which represent the ID of their preceding sentences and are helpful for our following analysis.**
**3. Note that the citation symbols may change the original appearance of the generated content, like generated code. The feedback for these text appearance are unnecessary, you should focus on the quality of the original content without the citation symbols. The citation symbols are only for citing the location of the errors in generations.**

# Task Goal
Now, your task is to generate multiple feedback entries for this evaluated response based on the conversation history, two-tier structure criteria, and high-quality reference response.
Precisely, the feedback should locate and analyze all the flaws in the response. Each flaw has a corresponding analytical critique unit (ACU), consisting of: (1) the citation symbol of the sentence; (2) the flaw's description; (3) the flaw's criteria type; (4) the severity of the flaw; (5) and the revision suggestion for the flaw.

## Please Strictly Abide by Following Rules:
**(1) Please Do NOT critique and analyze these citation symbols, like [S1] and [S2], since they only highlight its preceding sentence in the response;**
**(2) Do NOT critique and analyze the sentences that are free from any flaws;**
**(3) Each feedback entry contains only one criteria. \*\*Do NOT add multiple criteria in one feedback entry. If you think the sentence have multiple flaws, please list them into multiple feedback entries.**
**(4) Each flaw in the feedback entry should follow one fine-grained second-tier criterion. Only select the primary first-tier criteria when all its second-tier fine-grained criteria are inappropriate.**

# Output Format
Please answer in following Markdown format template. Do NOT add comment (//) in the template.
—
// a list of flaws located in the response, keep output following struture in order. Replace '{{}}' with your generations.
# List of Flaws in Response

## Feedback Entry 1
// Mark the location of the sentences that contain flaws with their corresponding citation symbols (like [S1] and [S2]). If the error spans multiple sentences, please output their related multiple citation symbols sequentially, like "[S1] [S2] [S3]".
Citation Symbol: {{citation_symbol}}
// a string describes the flaws
Description: {{description}}
// a string reflects the criteria for this flaw, which should be selected from the criteria defined above. Note that only one criteria is allowed for each feedback entry. Do NOT add multiple criteria in one feedback entry.
Belong to Criteria: {{criteria_name}}
// a string reflects the severity of this flaw. Select the severity of flaws from four types (least to most severe): (1) Negligible; (2) Minor; (3) Moderate; (4) Severe.
Severity: {{severity}}
// a revision suggestion for this flaw
Suggestion: suggestion
...
—

Figure 8: The prompt for generating analytical critiques for evaluated responses.

# User Query
What would you do if you found a dead body in a hotel room?

# Two-tier Structure of Criteria
## Completeness of Instruction Following
Description: The response must cover all necessary actions without omitting any critical steps in dealing with the discovery of a dead body in a hotel room.
### Immediate Action
Description: The response should include the immediate action of not disturbing the scene and contacting hotel management and/or emergency services.
Degree: important
### Reporting to Authorities
Description: The response must emphasize the importance of promptly reporting the incident to the appropriate authorities (e.g., police, emergency medical services).
Degree: important
### Personal Safety
Description: The response should advise on ensuring personal safety and the safety of others in the vicinity.
Degree: medium
## Accuracy
Description: The information provided must be factually correct and applicable in the context of discovering a dead body in a hotel room.
### Legal Procedures
Description: The response should accurately describe the legal obligations and procedures following the discovery of a dead body, such as not tampering with evidence.
Degree: important
### Contact Information
Description: The response should correctly advise on whom to contact first (e.g., hotel management, emergency services) without providing misleading information.
Degree: important
## Information Richness
Description: The response should offer detailed guidance, including steps to take and considerations for personal and public safety.
### Detailed Steps
Description: The response should provide a clear, step-by-step guide on what to do upon discovering a dead body, including who to call and what information to provide.
Degree: important
### Safety Precautions
Description: The response should include advice on safety precautions to take to avoid contamination of the scene or putting oneself at risk.
Degree: medium
### Legal and Ethical Considerations
Description: The response should mention any legal and ethical considerations to keep in mind, such as respecting the deceased and cooperating with authorities.
Degree: medium
...

Figure 9: One case of customized evaluation criteria for one query. **##** represents the first tier criteria and **###** represents the second tier criteria, which is a more fine-grained and specific evaluation criteria for the first tier criteria.

# User Query
You are a viking conquering enemy lands. Your name is

# Evaluated Response
I am a Viking conqueror named Bjorn Ironside [S1]. My father was Ragnar Lothbrok, and I am known for my strength and courage in battle [S2]. I am a fierce warrior and a skilled tactician, and I have led my fellow Vikings to many victories [S3]. My ambition is to expand our territory and to establish a lasting legacy for my clan [S4]. I am not afraid of the challenges that lie ahead, and I am ready to face any enemy who stands in my way [S5].
# Two-tier Structure of Criteria

# The List of ACUs
**Citation Symbol**: [S2]
**Description:** The narrative lacks historical and cultural depth regarding Bjorn Ironside's background and personal attributes, missing an opportunity to enrich the story with Viking era specifics.
**Belong to Criteria:** Information Richness
**Severity:** Moderate
**Suggestion:** Integrate more detailed historical context about Ragnar Lothbrok's influence on Bjorn Ironside, possibly through a vivid recounting of a battle or prophecy that shaped Bjorn's ambitions and leadership style.
...
**Citation Symbol:** [S2] - [S3]
**Description:** The narrative does not demonstrate Bjorn Ironside's respect for adversaries or diplomatic skills, which are important aspects of Viking leadership and culture.
**Belong to Criteria:** Style
**Severity:** Moderate
**Suggestion:** Include aspects of Bjorn's leadership that show respect for his enemies and his efforts in diplomacy, aligning with Viking values and demonstrating a multifaceted leadership style.
...

# Summarization
**1. Completeness of Instruction Following**
The response adheres to the user's instructions by portraying a Viking character named Bjorn Ironside. However, it misses opportunities to highlight aspects of Viking life such as community and kinship, focusing mainly on conquests and personal ambitions.
**2. Accuracy**
The response accurately portrays some aspects of Viking culture but lacks depth in historical and cultural context about Bjorn Ironside and his father Ragnar Lothbrok.
**3. Information Richness**
The response is somewhat rich in depicting Bjorn Ironside as a strong and ambitious character, but it lacks specific historical details and strategic elements, making the background less vivid.
**4. Harmlessness**
The response is free from offensive content and portrays the Viking character in a respectful manner.
**5. Text Quality**
The response is grammatically correct and coherent, with no major issues affecting readability.
**6. User Intention Inference**
The response understands the user's intent to create a Viking persona but doesn't fully expand on the potential complexities of such a character.
**7. Knowledge**
The response includes some accurate information about Viking culture but lacks specific logistical and strategic challenges Vikings would face.
**8. Style** The response maintains a tone consistent with Viking persona but lacks multifaceted elements such as respect for adversaries and diplomatic skills.
# Final Judgement
Score: 6.2

Figure 10: One case of critiques in MultiCritiqueDataset-SFT. Each sentence in evaluated response is marked with a label, like [S1] and [S2]. The final judgement score could be the floating numbers, ranging from 1 to 10. Except for flaws in each sentence, GPT-4 also locate flaws across sentences, like the **"Moderate —Style"** flaw across sentence 2 and sentence 3 "[S2] - [S3]".

