# OpenReview forum: "Training Language Models to Critique with Multi-Agent Feedback"
_ICLR.cc/2025/Conference — Submitted to ICLR 2025_

### Official Review · Reviewer_fcKr · 2024-11-02

**Soundness:** 2
**Presentation:** 3
**Contribution:** 2
**Rating:** 5
**Confidence:** 3

**Summary:**

The paper introduces MultiCritique, a data generation pipeline designed to enhance the critique ability of LLMs by using multi-agent feedback during both SFT and RL. MultiCritique aggregates feedback from multiple models, filtering out errors to produce a critique dataset. This method uses a structured approach where each model identifies specific flaws, followed by a meta-critique stage that refines the critiques for accuracy. The reinforcement learning stage further boosts critique quality using a preference-based dataset refined through multi-agent scoring of revision effectiveness. Experiments demonstrate that MultiCritique-trained models outperform existing models on benchmarks such as CRITICEVAL and CRITICBENCH.

**Strengths:**

The paper presents using multi-agent feedback through the MultiCritique pipeline, a deviation from previous single-model critique training methods. This generates critique data by aggregating feedback from multiple advanced models, thus raising the quality and robustness of the generated critiques.

This paper validates the model improvements on benchmarks CRITICEVAL and CRITICBENCH, with positive results. The inclusion of ablation studies demonsrates the contributions of each aspect of the proposed pipeline.

The authors provide a clear breakdown of the MultiCritique pipeline, detailing each stage from data preparation to the meta-critique and RL stages. The structured format of the ACUs is explained clearly, allowing readers to follow the critique generation and refinement processes.

**Weaknesses:**

The improvements demonstrated by the MultiCritique pipeline, while statistically significant, are relatively modest in scale, particularly in relation to the substantial computational complexity introduced by multi-agent / GPT4 feedback.

Additionally, the paper would benefit from exploring where the MultiCritique pipeline's improvements become saturated. By examining performance gains across a broader range of model sizes or critique tasks, the authors could better identify the conditions under which MultiCritique delivers more substantial gains.

**Questions:**

The multi-agent critique and meta-critique process likely incurs substantial computational expenses, especially given the high reliance on GPT-4 for critique refinement. A practical analysis of the computational cost relative to the quality gains would provide more context on the feasibility of scaling this approach for widespread or real-time applications.

---

> ### Author Response · Authors · 2024-11-20
>
> We really appreciate your valuable suggestions and insightful questions. Your questions are addressed as follows:
>
> ---
>
> **Q1/Q2: The improvements demonstrated by the MultiCritique pipeline, while statistically significant, are relatively modest in scale ...; The multi-agent critique and meta-critique process likely incurs substantial computational expenses, especially given the high reliance on GPT-4 for critique refinement. A practical analysis of the **computational cost relative to the quality gains** would provide more context on the feasibility of scaling this approach for widespread or real-time applications.**
>
> **A1/A2:** We appreciate the reviewer's concern about the computation cost related to performance. We would like to address this as below.
>
> While our approach does introduce additional computational overhead through multi-agent feedback and GPT-4 meta-critique, **our proposed MultiCritique significantly outperforms previous works in data efficiency.** As shown in Figure 2, our detailed analysis shows that MultiCritique achieves superior cost-effectiveness compared to existing approaches. With only 3K training samples (≈\\$890 in API costs), MultiCritique significantly outperforms baselines using 100K-257K samples (\\$1,915-\\$3,758 for UltraFeedback and Prometheus). **This represents a 2.15-4.22× improvement in data efficiency.** Since our current results already show state-of-the-art performance with modest computational resources, we didn't scale up the MultiCritique due to resource limitations.
>
> Additionally, we want to emphasize that our MultiCritique pipeline is primarily designed for offline dataset construction and optimization rather than real-time applications. The improved data quality and efficiency justify the one-time computational investment, as the MultiCritique dataset enables the improvement of efficient critique LLM's critique ability.
>
> We have supplemented these details in our revisions (Section 5.3).
>
> ---
>
> **Q3: Additionally, the paper would benefit from exploring where the MultiCritique pipeline's improvements become saturated. By examining performance gains across a broader range of model sizes or critique tasks, the authors could better identify the conditions under which MultiCritique delivers more substantial gains.**
>
> **A3:** We appreciate your valuable suggestions about the investigation of model sizes and critique tasks for MultiCritique. Let us clarify as below:
>
> 1. **Model Scale Analysis**
>
>    While our experiments on 7B-8B models are comprehensive, our investigation of larger models is insufficient. We have supplemented a new limitation section (Appendix A.5) to clarify this question. Our preliminary study demonstrates that the improvements brought by MultiCritique on the advanced Qwen2.5-72B-Instruct model are not very significant. Our analysis suggests two primary reasons for this phenomenon:
>
>    * These 70B models have been sufficiently fine-tuned, exhibiting very powerful critique and generalization ability. **For example, Qwen2.5-72B-Instruct achieves 83.62% F1 on CriticBench, surpassing GPT-4's 78.75%. Besides, its objective critique performance in CriticEval is also superior to GPT-4 (78.72% > 76.09%).** This high level of critique performance makes further improvements challenging.
>    * Our experiments on 70B models utilize only MultiCritique data for optimization. We suggest that the data mix strategy that integrates our dataset with other high-quality instruction data, might play a crucial role in improving their performance further.
>
>    **However, due to constraints in our computational resources and scheduling, our experiments and studies on these advanced 70B models are insufficient. We have incorporated these details into the limitation sections, and plan to address this limitation in our future work.**
>
> 2. **Critique Task Diversity**
>
>    We want to highlight that our chosen CriticEval and CriticBench benchmarks have encompassed a comprehensive range of critique tasks, including alignment, NLP, and diverse reasoning tasks. It enables the observations of saturation conditions of our work. To the best of our knowledge, these two benchmarks are comprehensive in evaluating critique abilities. Meanwhile, we have also elaborated the critique protocol limitations of our work in Appendix A.1, which will be solved in our future work.
>
> We appreciate the reviewer for bringing up these important points, which have helped us improve the quality of our paper.

---

> > ### Comment · Reviewer_fcKr · 2024-11-26
> > **Response to Author Comments**
> >
> > Thank you for your response. I maintain my original rating due to the substantial computational complexity introduced by multi-agent or GPT-4 feedback with modest performance gain, and this dataset construction method may not be innovative enough.

---

> > > ### Author Response · Authors · 2024-12-01
> > >
> > > Dear Reviewer fcKr,
> > >
> > > We hope this message finds you well. As we are approaching the final rebuttal deadline (December 2nd), we wanted to reach out respectfully to ensure that our previous responses have adequately addressed your concerns.
> > > Your feedback is invaluable to us, and we want to make sure we have fully understood and properly addressed all your questions. If you have any remaining concerns or require further clarification on any points, we would be more than happy to provide additional information.
> > >
> > > Thank you again for your time and effort in reviewing our work.
> > >
> > > Best regards,
> > >
> > > Authors of Submission1712

---

> ### Author Response · Authors · 2024-11-29
>
> We really appreciate your valuable feedback regarding the computational complexity and innovation aspects of our work. We would like to address these concerns with additional clarification and supporting evidence:
>
> ### **Regarding Performance Improvements and Computational Efficiency**
>
> As demonstrated in Section 5.3 and our previous evidence, MultiCritique represents a 2.15-4.22x improvement in data efficiency. **This observation demonstrates that MultiCritique achieves comparable or even better performance with 2.15-4.22x times less computation cost compared to strong baselines**. This also indicates that the quality of critique data is more crucial than quantity, justifying the additional computational cost during training for achieving superior critique capabilities.
>
> ### **Regarding Methodological Innovation**
>
> Unlike previous works, our work proposes **a unified multi-agent feedback framework that improves the critique ability of LLMs from both SFT and RL fine-tuning phases:**
>
> 1. **Before Critique (Section 3.1)**
> 	Our framework introduces three crucial information to **reduce the complexity of the critique task.**
>
> 2. **In the SFT phase (Section 3.2)**
> 	To address the bias of one single model, we propose to aggregate diverse and accurate critiques from multi-agents and mitigate the flawed critiques.
>
> 3. **In the RL phase (Section 3.3)**
> 	Beyond the behavior cloning of SFT, we propose to calibrate the preference critiques by utilizing the multi-agent feedback from the downstream revision task, facilitating the robust RL for improving LLM's critique ability.
>
> Experiments in Section 6 (Table 4) prove the effectiveness of our designs.
>
> ---
>
> Thank you again for your feedback, and we hope this detailed response could address your concerns. If you have other questions about our work, we would be more than happy to assist you.

---

### Official Review · Reviewer_CtUU · 2024-11-02

**Soundness:** 2
**Presentation:** 2
**Contribution:** 2
**Rating:** 5
**Confidence:** 4

**Summary:**

Overall, this paper is poorly written, describing certain motivations and methods that are difficult to understand.

> For example, in the abstract, the authors emphasize that the inherent complexity of model-generated critiques limits model performance, which they cite as a primary motivation for this work. However, MultiCritique originates from Multi-agent Feedback, which clearly adds to this complexity by involving multiple models.

**Strengths:**

The authors leverage multi-agent feedback to train models capable of critique, claiming that multi-agent feedback provides more accurate and critical information, which could address the inherent complexity of critique. However, the reviewer has expressed doubts about this claim. While the experimental setup is fairly robust, the descriptions of the experimental settings lack detail in most parts.

**Weaknesses:**

1. In terms of research on critique, other researchers have already explored similar ideas with LLMs. For instance, Jan Leike at OpenAI proposed "Self-critiquing models for assisting human evaluators," which this paper does not discuss. Additionally, while the authors discuss RLHF, they surprisingly omit foundational literature on RLHF, such as *Training language models to follow instructions with human feedback*. It appears that the authors did not thoroughly investigate relevant literature, which results in some contributions being overstated.

2. The terms “meta-cognitive” and “critique as a greater challenge than criticism” are ambiguous. Could the authors clarify these?

3. In Line 173, the authors state, "In total, we sample 10.7K queries covering 123 diverse task scenarios." However, the reviewer does not know what these 123 tasks entail or where to find this information in the appendix in this paragraph.

4. The authors use InternLM2-20B for scoring. How accurate is this reward model in the authors’ study, and how does it align with human evaluations at the three labeled response levels? Furthermore, Section 3.1 lacks a clear definition of these three levels.

5. In Section 4.1, the authors present Equation 1, where they condition \( Q_i \) and \( R_i \) to generate \( C_I \) and \( C \). How does this differ from the fine-tuning approach in the Aligner [3] paper, or are the two approaches identical?

6. In the experiments, I noticed that the authors applied SFT on InternLM2-7B-Chat-SFT, essentially layering SFT on a model that was already SFT-trained. How did they address potential issues of hallucination in this setup? Moreover, why did they not consider using Llama3-8B-Instruct as a base model?

7. This paper is based on multi-agent feedback, but it is unclear how the multi-agent approach is implemented. My understanding is that the feedback primarily comes from multiple language models. How does this approach differ from the multi-agent debate setups described in [4][5]? If the multi-agent approach in this paper merely relies on feedback from multiple LLMs without any designed communication between the agents, it appears to lack sufficient innovation compared to existing works on critique and debate. I recommend that the authors clarify this aspect.

---

### References:
[1] Self-critiquing models for assisting human evaluators
[2] Training language models to follow instructions with human feedback
[3] Aligner: Efficient Alignment by Learning to Correct
[4] Debating with More Persuasive LLMs Leads to More Truthful Answers
[5] Improving Factuality and Reasoning in Language Models through Multiagent Debate

**Questions:**

See above.

**Details Of Ethics Concerns:**

See above.

---

> ### Author Response · Authors · 2024-11-20
> **Responses to Reviewer (1/3)**
>
> We really appreciate your valuable suggestions and insightful questions. Your questions are addressed as follows:
>
> ---
>
> **Q1: ... the authors emphasize that the inherent complexity of model-generated critiques ... However, MultiCritique originates from Multi-agent Feedback, which clearly adds to this complexity by involving multiple models.**
>
> **A1:** There appears to be a misunderstanding regarding the relationship between task complexity and technique complexity. While our approach involves multiple models, it reduces the overall complexity of the critique task rather than increasing it. Let us clarify it in the following two aspects:
>
> 1. **Simplifying Critique Tasks with Crucial Information:**
>
>    As described in Section 3.1 (Crucial Information Collection), before generating critiques, our proposed MultiCritique specifically elicits three crucial information to reduce the complexity of critiques, including task description, customized evaluation criteria and reference response. Experimental results in Section 6 also prove the contributions.
>
> 2. **Improve Critiques with Multi-agent Feedback:**
>
>    The inherent complexity of the critique task makes it challenging for a single model to perform effectively, often resulting in flawed critiques that negatively impact downstream fine-tuned LLMs. Therefore, we propose a multi-agent feedback framework by aggregating diverse and accurate critiques from multi-agents. Besides, we propose to calibrate the preference critiques by utilizing the multi-agent feedback from the critique's downstream revision task, facilitating the robust RL for improving LLM's critique ability.
>
>    Extensive experimental results in Section 6 have proven the effectiveness of our designs.
>
> ---
>
> **Q2: In terms of research on critique, other researchers have already explored similar ideas with LLMs ... which results in some contributions being overstated.**
>
> **A2:** We sincerely appreciate the reviewer's valuable feedback on the literature review. While we acknowledge our oversight in not explicitly citing these fundamental works, we would like to respectfully clarify the distinct contributions of our work and demonstrate why we believe our claims are well-supported by evidence.
>
> #### **Novel Multi-Agent Feedback Framework for Critique Generation:**
>
> Although Saunders et al.'s work [1] on "Self-critiquing" pioneered the use of LLMs for critique, our approach fundamentally differs in several aspects:
>
> - We propose a novel multi-agent feedback mechanism that automatically collects diverse perspectives from multiple LLMs, without the need for costly human annotations.
> - We introduce crucial information (task description, evaluation criteria, and reference responses) to simplify the complex critique tasks, improving critique quality.
> - Our framework proposes a multi-agent feedback framework to aggregate diverse and accurate critiques from multi-agents and calibrate the preference critiques utilizing multi-agent feedback from downstream revision tasks, facilitating robust RL.
>
> #### **Automatic High-Quality Dataset Construction for Both SFT and RL:**
>
> Unlike previous RLHF works, like InstructGPT [2], our key innovation lies in:
>
> - Propose an automatic pipeline for generating high-quality critique datasets for both SFT and RL stages.
> - Propose Meta-Critique and Multi-Agent-Revision-Scoring mechanism to refine quality of critiques and preference critiques, facilitating robust improvement for SFT and RL.
>
> Our experimental results validate the positive contributions of our designs, showing that our 7B model achieves comparable performance to 70B models and approaches GPT-4 on CriticEval and CriticBench datasets. We have supplemented these two works in our revised manuscripts (Related Work) to better contextualize our work within the existing literature while maintaining our claims about its novel contributions.
>
> > [1] Self-critiquing models for assisting human evaluators
> >
> > [2] Training language models to follow instructions with human feedback

---

> ### Author Response · Authors · 2024-11-20
> **Responses to Reviewer (2/3)**
>
> **Q3: The terms “meta-cognitive” and “critique as a greater challenge than criticism” are ambiguous. Could the authors clarify these?**
>
> **A3:** We appreciate the reviewer's questions about these important concepts. Let us clarify them in detail:
>
> The concept of "meta-cognitive capability" encompasses higher-order cognitive processes that involve understanding and regulating one's thinking processes [3,4], extending well beyond mere critique. This capability comprises three essential components:
>
> - Self-understanding: the ability to comprehend its processes
> - Reflection: The capacity to evaluate responses against criteria
> - Self-regulation: The ability to adjust decisions (responses) based on reflection
>
> Our work covers two aspects:
>
> 1. Self-understanding: understand the task by introducing the crucial information, which also reduces the complexity of the critique task (Section 3.1)
> 2. Reflection: collect diverse and accurate critiques by utilizing multi-agent feedback and a Meta-Critique process (Section 3.2).
>
> We hope these clarifications help you better understand how our approach tackles the complex meta-cognitive aspects of critique generation, rather than merely producing surface-level criticism.
>
> > [3] Metacognition is all you need? using introspection in generative agents to improve goal-directed behavior
> >
> > [4] Metacognitive prompting improves understanding in large language models
>
> ---
>
> **Q4/Q5:  ... However, the reviewer does not know what these 123 tasks entail or where to find this information in the appendix in this paragraph ... Furthermore, Section 3.1 lacks a clear definition of these three levels.**
>
> **A4/A5:** We appreciate the reviewer's careful examination of our data preparation process. Let us address each point:
>
> #### **Tasks in MultiCritique:**
>
> We are sorry that we cited the wrong Section for the details about 123 tasks. The original text referenced Appendix C (last paragraph of Section 3.1), but the corresponding section should be Appendix D (Table 12). We have corrected this issue. Thank you very much for bringing this to our attention.
>
> #### **Response Quality Assessment with InternLM2-20B-Reward Scoring**
>
> Regarding InternLM2-20B for scoring, we want to clarify as following two points (motivation and performance):
>
> - **Motivation:** Our utilization of the reward model is to solve the uneven distribution of response quality in our dataset. Therefore, we utilize the reward model to coarse-grained judge the quality of responses and classify them into three performance durations (low, medium, and high). Thus, the accuracy of the reward model has little impact on our final collection of critiques.
> - **Performance of InternLM2-20B-Reward:** When we started our project, the InternLM2-20B-Reward was the top-tier reward model in the RewardBench leaderboard [5]. Besides, we explore the correlations between InternLM2-20B-Reward and human annotations at our work's earlier stage. Specifically, in discriminating between 200 pairs of high/low quality responses, InternLM2-20B-reward achieved a 95.3% consistency with human judgments, which is reliable to automatically evaluate the quality of responses. As described in Appendix C.1 (Table 7), these three quality levels are classified into low, medium, and high qualities, and the average quality scores are -1.41, 0.70, and 1.69, indicating a significant performance gap.
>
> We have enhanced Section 3.1 and extended Appendix C.1 to better highlight these important details while maintaining full documentation in the Appendix.
>
> > [5] Rewardbench: Evaluating reward models for language modeling
>
> ---
>
> **Q6: In Section 4.1, the authors present Equation 1 ... How does this differ from the fine-tuning approach in the Aligner ...**
>
> **A6:** Thanks for your question. While there might be superficial similarities in the mathematical formulation, our SFT fine-tuning differs from Aligner [6] in both **motivation and technique**.
>
> - **Motivation:** Aligner [6] focuses on optimizing response revision upon one upstream LLM, while our approach is specifically designed to enhance LLMs' critique abilities rather than revision capabilities.
> - **Technique:** In Equation 1, we jointly optimize the crucial information and critiques in a unified framework, aiming to enhance the understanding of query-response, which is theoretically grounded in human metacognition [1,2]. This joint optimization approach differs significantly from Aligner's optimization.
>
> **While Aligner doesn't align with our topic of critique, we add it into the related work (line 161 in related work) since it can be viewed as a potential multi-agent solution that integrates upstream and downstream models. Thanks for your suggestion.**
>
> > [6] Aligner: Efficient Alignment by Learning to Correct

---

> ### Author Response · Authors · 2024-11-20
> **Responses to Reviewer (3/3)**
>
> **Q7: ... authors applied SFT on InternLM2-7B-Chat-SFT ... How did they address potential issues of hallucination? Moreover, why did they not consider using Llama3-8B-Instruct as a base model?**
>
> **A7:** Thank you for these important questions. Let us address them separately.
>
> Firstly, our preliminary experiments and human annotations have not observed any significant hallucination issues in critiques introduced by the additional SFT process. However, we observed error amplification problems when fine-tuning on GPT-4's flawed samples during the SFT phase.
>
> Regarding the choice of the base model, our RL implementation is not designed for the Llama3 series due to its late publication date (April 2024, our project starts in March 2024). Following your suggestions, we have conducted MultiCritique-SFT fine-tuning on three additional advanced base models: InternLM2.5-7B-Chat, Llama-3-8B-instruct and Qwen2.5-7B-Instruct, detailed in our revised manuscripts **(Appendix H)**. Experimental results show that even though these advanced LLMs have already demonstrated very strong zero-shot critique capabilities, our MultiCritique can further improve their critique ability. For example, the average improvement of these three additional models in CriticBench is 15.9%.
>
> ---
>
> **Q8:  ...it appears to lack sufficient innovation compared to existing works on critique and debate. I recommend that the authors clarify this aspect.**
>
> **A8:** We want to emphasize that the implementations of our proposed multi-agent feedback have been described as the core contributions in Section 3.2 and Section 3.3.
>
> We want to highlight that most previous multi-agent frameworks aim to improve the LLM's alignment and response quality, while our work aims to improve the critique ability of LLMs with multi-agent feedback through both the SFT and RL stages, rather than the alignment refinement. We will clarify more details in as below:
>
> #### **MultiCritique-SFT Stage**
>
> First, as described in Section 3.2, our proposed MutiCritique-SFT framework consists of three steps, differing from previous multi-agent debate approaches [7,8]:
>
> 1. **Multi-agent Analytical Critique:** multiple advanced LLMs independently generate Analytical Critique Units (ACUs). Each ACU is carefully designed to identify and address specific flaws in the evaluated response, consisting of five key values, such as error descriptions and suggestions.
> 2. **Meta-Critique Classification:** We introduce a meta-critique mechanism that evaluates the quality of each ACU through classification into human-annotated quality categories, given all the ACUs generated by multi-agent.
> 3. **Critique Summarization:** Rather than relying on debate outcomes, we employ critique summarization to consolidate diverse perspectives while filtering out flawed critiques. This approach preserves the unique insights of each agent while ensuring critique quality.
>
> **Regarding your question, Meta-Critique, and Critique Summarization can be viewed as a special communication mechanism among agents designed for the critique task.** Besides, as described in Section 3.2 (second paragraph), we have already explained why we did not adopt the traditional multi-agent debate approach. Specifically, we observed that multi-agent debate reduces the diversity of critiques due to the complexity of critique tasks, which harms the diversity and accuracy of critiques.
>
> #### **MultiCritique-RL Stage**
>
> We calibrate to refine the preference critiques with the multi-agent feedback from downstream revisions, facilitating robust RL during fine-tuning.
>
> We hope these distinctions demonstrate the innovation of our work compared to existing multi-agent debate and critique works. These details have been supplemented into the related work in our revised submission. Thanks for your questions.
>
> > [7] Debating with More Persuasive LLMs Leads to More Truthful Answers
> >
> > [8] Improving Factuality and Reasoning in Language Models through Multiagent Debate

---

> > ### Comment · Reviewer_CtUU · 2024-11-23
> > **comment by Reviewer**
> >
> > Thank you for your reply. I will review the response carefully.

---

> ### Author Response · Authors · 2024-11-24
> **Thanks for your response**
>
> We sincerely appreciate your response and thank you for taking the time to review our responses. We look forward to engaging in a constructive discussion about our work.

---

> ### Author Response · Authors · 2024-11-29
>
> **We really appreciate your valuable feedback and your time and effort in reviewing other reviewer's comments.** We would like to address your concerns regarding computational complexity and innovation as follows:
>
> ### **Regarding Computational Cost**
>
> As demonstrated in Figure 2 and Section 5.3, MultiCritique represents a 2.15-4.22x improvement in data efficiency. **This observation demonstrates that MultiCritique achieves comparable or even better performance with 2.15-4.22x times less computation cost compared to strong baselines, like Auto-J, UltraFeedback and Prometheus**. This also indicates that the quality of critique data is more crucial than quantity, justifying the additional computational cost during training for achieving superior critique capabilities.
>
> ### **Regarding the Innovation in the training process and dataset collection**
>
> **Our work primarily focuses on designing an effective multi-agent feedback framework to enhance the LLM's critique ability. Therefore, we don't modify the training loss function and training process, which is not the objective of our work.**
>
> The innovations of our work centers on a unified multi-agent feedback framework that enhances LLM's critique capabilities through both supervised fine-tuning (SFT) and reinforcement learning (RL) phases, effectively addressing two fundamental challenges in this domain. We have elaborated on these innovations in detail in our previous response **(Q8/A8)** and the **"Global Response - Novelty"** section.
>
> ---
>
> We hope our clarification can address your concerns. If you have other questions about our previous comments, we would be more than happy to assist you.

---

### Official Review · Reviewer_X97P · 2024-11-02

**Soundness:** 3
**Presentation:** 3
**Contribution:** 3
**Rating:** 6
**Confidence:** 3

**Summary:**

The paper proposes MultiCritique, a novel data generation pipeline aimed at enhancing language models' (LLMs) critique abilities by leveraging multi-agent feedback during supervised fine-tuning (SFT) and reinforcement learning (RL) stages. The proposed MultiCritique pipeline aggregates critiques from multiple advanced LLMs, utilizing key meta-critique processes to classify and consolidate high-quality feedback. This results in the construction of a new MultiCritiqueDataset, which shows empirical improvements on CRITICEVAL and CRITICBENCH benchmarks, demonstrating superior performance in critiquing abilities. Notably, the approach effectively competes with models up to ten times its size, approaching the critique capabilities of models like GPT-4.

**Strengths:**

1. The paper introduces a multi-agent framework that significantly improves critique dataset quality by aggregating and refining feedback from diverse LLMs. The structured use of Analytical Critique Units (ACUs) and meta-critique classification addresses the limitations of single-agent feedback and provides a robust critique foundation. The structured use of Analytical Critique Units (ACUs) and meta-critique classification addresses the limitations of single-agent feedback and provides a robust critique foundation.

2. The MultiCritiqueDataset improves LLM critique ability considerably over traditional supervised datasets, evidenced by impressive gains on CRITICEVAL and CRITICBENCH

3. The paper has quite extensive evaluation and analysis that support the method.

**Weaknesses:**

See question.

**Questions:**

1. While the pipeline effectively aggregates critiques, the reliance on multiple LLMs (including GPT-4) for meta-critique classification may introduce considerable computational and financial overhead. Could you provide quantitative result and  a discussion on efficiency, potential trade-offs, or future work focused on optimizing this aspect?

2. Although the multi-agent framework is well-justified, could you provide a more explicit differentiation from other multi-agent critique frameworks (e.g., Arena learning and Stable Alignment). Clearer explanations on how MultiCritique uniquely contributes to critique generation and feedback quality.

---

> ### Author Response · Authors · 2024-11-20
>
> We really appreciate your valuable suggestions and insightful questions. Your questions are addressed as follows:
>
> ---
>
> **Q1: While the pipeline effectively aggregates critiques, the reliance on multiple LLMs (including GPT-4) for meta-critique classification may introduce considerable computational and financial overhead. Could you provide quantitative result and a discussion on efficiency, potential trade-offs, or future work focused on optimizing this aspect?**
>
> **A1:** We sincerely appreciate the reviewer's insightful questions regarding the efficiency and trade-offs of our proposed MultiCritique framework. We answer your questions as below:
>
> 1. **Data Efficiency Analysis**
>
>    As shown in Figure 2 in Section 5.3, our detailed analysis shows that MultiCritique achieves superior cost-effectiveness compared to existing approaches. With only 3K training samples (≈\\$890), MultiCritique significantly outperforms baselines using 100K-257K samples (\\$1,915-\\$3,758 for UltraFeedback and Prometheus). **This represents a 2.15-4.22× improvement in data efficiency.**
>
> 2. **Future Optimization Directions**
>
>    We acknowledge the computational overhead of MultiCritique, and future works could focus on following aspects to improve it:
>
>    1. Investigating multiple samller LLMs for meta-critique, like PoLL [1]
>    2. Distilling meta-critique ability from GPT-4 to efficient LLMs.
>
> We have extended Section 5.3 to include these details. Thanks for your suggestions.
>
> > [1] Replacing judges with juries: Evaluating llm generations with a panel of diverse models
>
> ---
>
> **Q2: Although the multi-agent framework is well-justified, could you provide a more explicit differentiation from other multi-agent critique frameworks (e.g., Arena learning and Stable Alignment). Clearer explanations on how MultiCritique uniquely contributes to critique generation and feedback quality.**
>
> **A2:** We appreciate your questions regarding the differences between MultiCritique and previous works. **The main differentiation lies in the targets and techniques.** Let us clarify as below:
>
> As described in our related work, current multi-agent frameworks are widely used in two applications: (1) multi-agent framework for LLM alignment; (2) multi-agent framework for LLM-based evaluation.
>
> #### **Multi-agent framework for LLM alignment**
>
> Numerous works prove the effectiveness of multi-agent framework for improving LLM's alignment and response quality. For example, Arena Learning leverages a Llama-3-70B model as a judge to evaluate battle results between multiple models for target LLM improvement. Similarly, Stable Alignment employs multi-agent critiques for alignment refinement. Unlike these works, our work propose a unified multi-agent framework to improve LLM's critique ability, rather than LLM's alignment.
>
> #### **Multi-agent framework for LLM-based evaluation**
>
> Multi-agent frameworks are also used to improve the reliability of LLM-based evaluations, such as ChatEval and PoLL (described in related work). While these works improves the evaluation quality during the inference stage, our work focuses primarily on improving the critique ability of LLMs through the SFT and RL fine-tuning, aiming to overcome two challenges in recent critique research: (1) limited critique dataset quality by single model bias; and (2) upper bound brought by mainstream critique SFT approaches.
>
> More detailed clarifications are placed in **"Global - Novelty". We also extend the "Multi-agent Framework" paragraph in our related work to better clarify our differences with previous works.** Thanks for your suggestions.

---

> > ### Comment · Reviewer_X97P · 2024-11-25
> > **Response to the authors**
> >
> > Thank you for your response. The author has addressed my concern. I will maintain my original score.

---

> ### Author Response · Authors · 2024-11-25
>
> We sincerely appreciate your feedback and are pleased that we have successfully addressed your concerns. If you have any additional questions or require further clarification, we would be more than happy to assist you.

---

### Official Review · Reviewer_3oRZ · 2024-11-03

**Soundness:** 3
**Presentation:** 4
**Contribution:** 3
**Rating:** 6
**Confidence:** 3

**Summary:**

The paper, tackles the inherent difficulty in enhancing critique capabilities within large language models (LLMs). Critique ability, a key aspect of human meta-cognition, is essential for LLMs as it allows them to identify and address flaws in responses. This skill is pivotal for reliable automatic evaluation and model self-improvement. However, training models to critique effectively poses challenges. Most existing methods rely on supervised fine-tuning (SFT) using critiques generated by a single LLM, typically a model like GPT-4. Although effective to a degree, these single-model critiques can propagate the original model’s biases and limitations, leading to inaccuracies and diminished critique quality.

To address these issues, the authors introduce a novel data generation pipeline, MultiCritique, that uses multi-agent feedback to enhance the quality and diversity of critiques. Rather than depending on a single model’s perspective, MultiCritique aggregates feedback from multiple advanced LLMs, allowing for a richer, more balanced dataset. This multi-agent approach is implemented in both the SFT and reinforcement learning (RL) stages, producing a dataset called MultiCritiqueDataset. With this dataset, the model undergoes fine-tuning and then reinforcement learning, where preference scoring further refines the critique quality based on feedback from several agents.

The study’s experiments show that a 7B model fine-tuned with the MultiCritiqueDataset significantly outperforms other models in critique tasks, even rivaling the performance of much larger, closed-source models like GPT-4. The authors argue that the multi-agent approach not only yields a higher-quality critique dataset but also preserves feedback diversity, making it a robust solution for training LLMs in critique ability. This advancement offers an effective pathway toward more self-improving, reliable, and critique-capable language models

**Strengths:**

- overall addressing an important problem of developing small high qulaity critique model
- new multi-agent feedbakc approach to curate dataset
- creation of a new high qulaity multi-critique dataset, incorporating diverse queries,and crucial information collection
- naturla integration and use of RL
- Strong experimental valiadtion, showing 7b FT models perform at par or better with closed source and 70b model on criticeval and critic bench

**Weaknesses:**

- not very technically novel, looks like a fairly stratightforward principle of ensembling for dataset collection
- Performance is still limited by an upperbound of the the base LLMs, and it is not clear if we could sclae to 70b, would the performance be better

**Questions:**

- No major questions, I think the paper ideas a straightforward and easy to follow, the presentation is good, and the ideas are well executed
- From a research perspective, I think novelty is lacking a bit, neverthless it is an important contribution

---

> ### Author Response · Authors · 2024-11-20
>
> We really appreciate your valuable suggestions and insightful questions. Your questions are addressed as follows:
>
> ---
>
> **Q1/Q2: ... I think the paper ideas a straightforward and easy to follow ... ensembling for dataset collection ...; From a research perspective, I think novelty is lacking a bit, neverthless it is an important contribution**
>
> **A1:** We are delighted that you acknowledge our presentations and contributions. We would like to emphasize that our work is not an ensemble for dataset collection. To overcome two challenges of critique research field, i.e., flawed critiques of one single model and upper bound of SFT fine-tuning, we propose a multi-agent feedback framework that aggregates diverse and accurate critiques from multi-agents, exhibiting superior quality than each model's critiques (as evidenced in Section 6, Table 4). Furthermore, beyond the merely behavior cloning in SFT, we propose to calibrate the preference critiques by utilizing the multi-agent feedback from the critique's downstream revision task, facilitating the robust RL for improving LLM's critique ability.
>
> You can find more clarification about out innovation in **"Global Response - Novelty"**.
>
> ---
>
> **Q3: Performance is still limited by an upperbound of the the base LLM .. if sclae to 70b, would the performance be better**
>
> **A3:** We appreciate this insightful question. While our experiments on 7B-8B models are comprehensive, our investigation of larger models is insufficient. We have supplemented a new limitation section (Appendix A.5) to clarify this question. Our preliminary study demonstrates that the improvements brought by MultiCritique on the advanced Qwen2.5-72B-Instruct model are not very significant. Our analysis suggests two primary reasons for this phenomenon:
>
> - These 70B models have been sufficiently fine-tuned, exhibiting very powerful critique and generalization ability. **For example, Qwen2.5-72B-Instruct achieves 83.62% F1 on CriticBench, surpassing GPT-4's 78.75%. Besides, its objective critique performance in CriticEval is also superior to GPT-4 (78.72% > 76.09%).** This high level of critique performance makes further improvements challenging.
> - Our experiments on 70B models utilize only MultiCritique data for optimization. We suggest that the data mix strategy that integrates our dataset with other high-quality instruction data, might play a crucial role in improving their performance further.
>
> **However, due to constraints in our computational resources and scheduling, our experiments and studies on these advanced 70B models are insufficient. We have incorporated these details into the limitation sections, and plan to address this limitation in our future work.**

---

### Official Review · Reviewer_zQhf · 2024-11-04

**Soundness:** 3
**Presentation:** 3
**Contribution:** 2
**Rating:** 6
**Confidence:** 3

**Summary:**

This paper proposes a data generation pipeline utilising some of the SoTA LLMs (OpenAI, Claude, Qwen etc) both for SFT and RL tuning in order to improve the critiquing ability of LLMs. The authors experiment on CriticEval and CriticBench, where their SFT and RL datasets provides a good boost in performance.

**Strengths:**

- It is a Practically useful work
- Good improvement in results in CriticEval and CriticBench benchmarks

**Weaknesses:**

- Mostly this paper feels like just a bit of a formalisation for distilling “critiquing” ability from GPT-4 by generating critiquing data both for SFT and RL tuning. The data generation process is also quite heavily dependent on GPT-4 abilities.
- Novelty of this work is not very clear to me

**Questions:**

- What is the generalizabilty of this kind of fine-tuning of the critiquing ability. Can the authors show whether finetuning on critiquing dataset generated from general alignment datasets (like OpenHermes, OpenAssistant) alone can still help in generating critiques in unseen domains like math and code.
- The performance improvement of the MultiCritiqueDataset-RL over MultiCritiqueDataset-SFT is not very clear - for some datasets there seems to be improvement while for others it is hurting or improvement is negligible. Can the authors comment on why this happening? Also what about including both MultiCritiqueDataset -SFT and -RL?
- Minor concern about terminology - I am not quite sure why the authors stress on “agent” feedback — this work on critique generation seems like yet another instruction tuning task, this in general can be helpful in agentic tasks but I don’t see why this is termed “multi-agent”

---

> ### Author Response · Authors · 2024-11-20
>
> We really appreciate your valuable suggestions and insightful questions. Your questions are addressed as follows:
>
> ---
>
> **Q1/Q2: Mostly this paper feels like just a bit of a formalisation for distilling “critiquing” ability from GPT-4 ...; Novelty of this work is not very clear to me**
>
> **A1/A2:** We appreciate the reviewer's question. We would like to clarify that our work does not distill GPT-4's critique capabilities. Instead, we propose a multi-agent feedback framework that aggregates diverse and accurate critiques from multi-agents, which exhibit superior quality than each model's critiques (as evidenced in Section 6, Table 4). Furthermore, we calibrate the preference critiques by utilizing the multi-agent feedback from the critique's downstream revision task, facilitating the robust RL for improving LLM's critique ability (Section 3.3). **This preference critique calibration process does not rely on GPT-4.**
>
> Overall, our work does not distill the critique ability of GPT-4. Please refer to **"Global Response - Novelty", and "Global Response - GPT-4 dependency"** for more clarifications.
>
> ---
>
> **Q3: What is the generalizabilty of this kind of fine-tuning ... can still help in generating critiques in unseen domains like math and code.**
>
> **A3:** We appreciate your important question about the generalization capability of our approach. Our extensive experiments demonstrate a strong generalization of MultiCritique to unseen domains, supported by the following evidence:
>
> 1. **Strong Improvement on Unseen Domains in CriticBench**
>
>    **Symbolic and Algorithm reasoning tasks in CriticBench are explicitly excluded from our training data (see Table 12, Appendix D), making them out-of-domain evaluation tasks.** Table 1 demonstrates that MultiCritiqueDataset-SFT significantly improves the critique ability on symbolic  (from 18.82% to 57.04%) and algorithm reasoning (from 14.29% to 51.85) tasks. Besides, RL further improves the critique ability on these two unseen tasks.
>
> 2. **Ablation Study of Removing Math and Code Tasks**
>
>    Following your suggestion, we also supplement the ablation study to study the generalization of MultiCritique to math and code domains. Specifically, we exclude the math and code training samples in MultiCritiqueDataset-SFT, and evaluate the fine-tuned model's generalization on these two unseen tasks. Partial results in CriticBench are shown as follows, and complete results are shown in our revised submission **(Generalization to Unseen Tasks in Section 6)**:
>
>    | Models | Math | Comm. | Symb. | Algo. | Code | Overall |
>    | - | - | - | - | - | - | - |
>    |InternLM2.5-7B-Chat | 59.46 | 48.26 | 42.63 | 37.21 | 63.97 | 53.98|
>    | + MultiCritique w/o Math&Code | **88.44** | **60.42** | **55.02** | **45.91** | **77.63** | **71.78** |
>
>    Results demonstrate that model fine-tuned on dataset without math and code samples can still achieve significant improvements on these two tasks.
>
> ---
>
> **Q4: The performance improvement of the MultiCritiqueDataset-RL over MultiCritiqueDataset-SFT is not very clear for some tasks ... Also what about including both MultiCritiqueDataset -SFT and -RL?**
>
> **A4:** We appreciate the reviewer's insightful questions about the performance differences between MultiCritiqueDataset-RL and MultiCritiqueDataset-SFT. Let us address these points comprehensively:
>
> 1. **Performance Variations Across Different Tasks**
>
>    Table 1 demonstrates that the improvement brought by MultiCritique-RL on mathematics, coding, and commonsense reasoning tasks in the CriticBench benchmark is modest. Since they are covered in MultiCritique, SFT's improvements are already significant. As a result, RL may not necessarily bring sufficient improvements. On the contrary, on unseen tasks, RL could bring substantial improvements (57.04% to 61.51% on Symbolic, and 51.85% to 57.76% on Algorithm).
>
> 2. **Rationale for Separate SFT and RL Phases**
>
>    Following the established RLHF workflow, like InstructGPT [1], we adopt the typical two-phase learning approach (SFT followed by RL), where SFT first calibrates the basic output behavior, followed by RL for further alignment.
>
> > [1] Training language models to follow instructions with human feedback
>
> ---
>
> **Q5: Minor concern about terminology ... but I don’t see why this is termed “multi-agent”**
>
> **A5:** Thanks for your question. Following previous works [2], the term "multi-agent feedback" is chosen to accurately reflect the distinct characteristics of our MultiCritique, aligning with previous works in multi-agent frameworks (described in related work). Unlike traditional instruction tuning that relies on a single model's output, our approach fosters diverse thinking of multiple models on critiquing response quality, overcoming the challenge of flawed critiques of one single model.
>
> > [2] ChatEval: Towards Better LLM-based Evaluators through Multi-Agent Debate

---

> > ### Comment · Reviewer_zQhf · 2024-11-26
> >
> > Thank you for your responses. While the additional experiments on unseen domains is interesting, i feel the main contribution of this work is the construction of the multicritique dataset. Based on that i would like to maintain my original score.

---

> ### Author Response · Authors · 2024-11-29
>
> **We really sincerely appreciate your positive feedback on our unseen domain experiments.** We would like to take this opportunity to further clarify the comprehensive nature of our work's main contribution.
>
> **Our primary contribution lies in developing MultiCritique, a unified multi-agent feedback framework that addresses two fundamental challenges in critique research from both SFT and RL fine-tuning phases**: (1) The bias of single-model critiques; (2) The performance upper bound by conducting supervised fine-tuning on critiques.
>
> **The construction of the MultiCritiqueDataset (both SFT and RL parts) is an integral extension and implementation of the MultiCritique framework.** The superior performance of models trained on these datasets validates the effectiveness of the MultiCritique framework.
>
> We hope this clarification illustrates how our work's contribution extends beyond dataset construction to encompass a comprehensive solution for improving LLMs' critique abilities.
>
> Thank you again for your valuable feedback.

---

### Official Review · Reviewer_rDX7 · 2024-11-05

**Soundness:** 3
**Presentation:** 3
**Contribution:** 2
**Rating:** 8
**Confidence:** 2

**Summary:**

This paper proposes a critique data generation pipeline called MultiCritique to enhance the critique capabilities of language models. The process is divided into three steps. In Step 1, the authors selected queries from six datasets and used 11 language models to generate responses. These responses were rated by InternLM2-20B-reward into three quality levels, and GPT-4 was used to extract and summarize crucial information (CI) for each query-response pair. In Step 2, the query-response-CI tuples were input into multiple language models to generate aspect-specific critiques, referred to as Analytical Critique Units (ACUs). GPT-4 then classified these ACUs into seven quality categories and aggregated them into a final critique summary, which could be used for SFT of language models. In Step 3, the ACUs generated in Step 2 were used to construct preference pairs, which were then employed for reinforcement learning fine-tuning using PPO.
The evaluation of this pipeline was conducted on two benchmarks: CriticEval (unpublished) and CriticBench (ACL Findings '24).

**Strengths:**

- The paper is well-written and effectively presented, with technical details that are clearly explained and easy to follow. Figure 1 is particularly effective in illustrating the entire MultiCritique pipeline.
- The evaluations and ablation studies are comprehensive, providing a thorough analysis of the model's performance and the impact of various components.
- The MultiCritique SFT dataset, if made open-source, could be a good contribution to the field due to its careful construction aimed at reducing variances and biases.
- The limitations section is well-prepared, addressing most concerns that might arise from the study.

**Weaknesses:**

- Dependence on GPT-4: The project heavily relies on GPT-4 for classifying and summarizing critiques, which may introduce biases into the final critique summaries. It is not hard to say that the entire project is feasible because of the existence of GPT-4. This reliance potentially undermines the benefits of using multiple agents in earlier steps. I know it might be a bit too much to ask, but it would be interesting to explore the impact of using a different model, such as Claude 3.5 Sonnet, to see how the results might differ.
- Specificity Towards Critique Ability: The model's focus on improving critique ability raises concerns about overfitting to these specific tasks. It would be beneficial to assess whether this specialization affects other capabilities, such as chat performance. Evaluating the model on benchmarks like AlpacaFarm Arena-Hard could provide insights into any potential degradation in general capabilities.

**Questions:**

- Table 4 is somewhat unclear to me. Could you elaborate on what was specifically done in this ablation study? Are you fine-tuning from the same baseline models and then using different datasets, such as MultiCritique-SFT versus critiques generated solely by GPT-4-turbo and other individual models? Additionally, are these individual critiques extracted from the MultiCritique dataset, or are they generated separately?

---

> ### Author Response · Authors · 2024-11-20
>
> We really appreciate your valuable suggestions and insightful questions. Your questions are addressed as follows:
>
> ---
>
> **Q1: Dependence on GPT-4: The project heavily relies on GPT-4 ... such as Claude 3.5 Sonnet, to see how the results might differ.**
>
> **A1:** We appreciate your valuable questions about our proposed method. We want to emphasize that our MultiCritique is a general and model-agnostic framework, and GPT-4 serves only as one possible meta-critique judge model. Following your suggestions, we utilize advanced Claude-3.5-sonnet and Qwen2.5-72B-Instruct to conduct the Meta-Critique and Critique Summarization (Section 3.2) and analyze their correlations with GPT-4. Our analysis reveals strong correlations between these models and GPT-4, confirming their viability as robust alternatives within our framework. **You can find more details in "Global Response - GPT-4 Dependency" and Appendix G in our revised manuscript.**
>
> ---
>
> **Q2: The model's focus on improving critique ability ... Evaluating ... like AlpacaFarm Arena-Hard ... degradation in general capabilities.**
>
> **A2:** We appreciate your valuable question about potential overfitting to critique tasks. Although solely fine-tuning on MultiCritique might lead to degraded performance in general ability, MultiCritique could be integrated into the existing SFT dataset pool to overcome this issue. Our experimental results prove that MultiCritique enhance both critique ability and general capabilities of LLMs.
>
> Specifically, we integrate MultiCritique-SFT into open-source instruction-tuning datasets and optimize the InternLM2-7B-Chat-SFT model. Then, we evaluate both the critique and general capabilities of the LLM. The evaluation of general capabilities is split into two aspects:
>
> 1. **Objective evaluation** computes the average performance on 18 famous benchmarks, like MMLU and GSM8K.
> 2. **Subjective evaluation** uses famous CompassJudger toolkit [1] to evaluate model performance on 4 general alignment benchmarks.
>
> Partial results are shown as below, and complete results are shown **in Section 6 and Appendix C (Table 10) of our revised submission.**
>
> | -    | CriticBench    | Average Objective Score|AlignBench    | AlpacaEval   | AlpacaHard    | MTBench101    |
> | ---- | ----|- | ---- | ---- | ---- | ---- |
> | w/o MultiCritique     | 38.8     |55.23 |5.07     | 23.51     | 22.78     |    7.75  |
> | w/ MultiCritique |**71.6**|**55.43**|**5.1**|**27.43**|**23.28**|**7.81**|
>
> The results demonstrate that model optimized on our integrated dataset achieves consistent improvements in both critique and general capabilities.
>
> > [1] CompassJudger-1: All-in-one Judge Model Helps Model Evaluation and Evolution
>
> ---
>
> **Q3: Table 4 is somewhat unclear to me ...**
>
> **A3:** We appreciate this important question. Let us clarify the motivation and experimental setup in detail:
>
> ### **Motivation**
>
> This ablation study aims to systematically evaluate the effectiveness of our MultiCritique-SFT pipeline compared to individual model feedback that is widely used in previous works. Specifically, we investigate whether the aggregated multi-agent feedback outperforms each one-single-model's individual critiques.
>
> ### **Implementation Details**
>
> 1. **Base Model:** InternLM2-7B-Chat-SFT, the same as the main experiment.
> 2. **Dataset Format:** The dataset in this ablation study is slightly different from that in the main experiment (Table 1), consisting of two parts:
>    1. **Distinct parts:** crucial information and analytical critiques (a list of ACUs), without the summarization and judgment score.
>       1.  **MultiCritique-SFT Critiques:** Critiques are generated through our MultiCritique-SFT pipeline, which aggregates accurate ACUs from multiple models via meta-critique classification.
>       2. **Four Individual LLMs Critiques:** Critiques are extracted from our **raw** MultiCritique-SFT dataset, using feedback generated independently by four models.
>    2. **Shared parts:** to enable the objective evaluation on CriticEval and CriticBench benchmarks, we supplement 5% of samples from MultiCritique-SFT to **ensure the fair comparison**, which consists of crucial information, analytical critiques, summarization and judgment score.
>
> ### **Experimental Results**
>
> Table 4 demonstrates that MultiCritique-SFT achieves substantial improvements over individual model feedback, validating that our multi-agent feedback framework effectively aggregates the strengths of multiple models while mitigating their individual limitations.
>
> We have incorporated these details in our revision to better illustrate the experimental design and findings. Thank you for helping us improve the paper's clarity.

---

> > ### Comment · Reviewer_rDX7 · 2024-11-26
> > **Thank you**
> >
> > Thank you for the efforts you spent on the rebuttal. I have decided to maintain my score.

---

> > > ### Author Response · Authors · 2024-11-29
> > >
> > > We sincerely appreciate your careful review of our paper and rebuttal. We are grateful for your recognition of our contributions in improving the critique ability of LLMs through our proposed MultiCritique framework.
> > >
> > > Thank you again for your time and valuable feedback throughout the review process.

---

### Author Response · Authors · 2024-11-20
**Global Response**

We thank all the reviewers for their helpful comments and suggestions. Overall, we are delighted that reviewers found our work is "clear and well-written" (Reviewer rDX7, fcKr, 3oRZ), and our contributions and improvements are "significant" (Reviewer rDX7, zQhf, 3oRZ, X97P, fcKR). Besides, we are also delighted that all reviewers thought our experiments are "comphrensive and robust".

Following suggestions of reviewers, we have updated our submission with additional experimental results showing the generalization to general capabilities (Section 6 "MultiCritique Improves the General Capability of LLMs"), generalization to unseen tasks (Section 6 "Generalization to Unseen Tasks"), and results on more base LLMs (Appendix H). We also re-written some paragraphs in our main text and Appendix for answering some reviewer questions. **All the revisions in PDF are highlighted in blue. In the following content, we will answer some common questions of reviewers.**

---

## **Novelty**

We appreciate the comments of Reviewer zQhf and 3oRZ regarding the novelty of our work. We will clarify as following aspects:

### **Fundamental Challenges and Limitations in Prior Work**

We want to highlight the challenges in previous works that we are trying to solve.

- **Inherent Complexity:** Critique requires LLMs to understand the tasks, and then provide constructive feedback on response, making it challenging even for advanced models like GPT-4.
- **Bias of One Single Model:** Previous critique-tuning approaches rely on single-model (GPT-4) critiques for SFT. When the teacher model (GPT-4) makes errors, these flaws are propagated to the student model during fine-tuning.
- **Upper bound Limitation of SFT:** The reliance on SFT creates an inherent upper bound on critique performance.

### **Distinct Targets and Task Formulation**

As described in related work, most previous multi-agent works primarily focus on improving LLM's alignment and response quality. While there are some multi-agent works that improve the LLM-based evaluation during the inference stage, our motivation is to design a unified multi-agent feedback framework to improve LLM's critique ability through both the SFT and RL fine-tuning stages.

### **Technical Contributions**

To address these challenges, our work introduces three solutions:

- **Introducing Crucial Information:** Our framework introduces three crucial information to **reduce the complexity of the critique task.**
- **MultiCritique-SFT Framework:** To address the bias of one single model, we propose to aggregate diverse and accurate critiques from multi-agents and mitigate the flawed critiques. **Experiments in Section 6 (Table 4) demonstrate that our critiques is better than each model's critique.**
- **MultiCritique-RL framework:** Beyond the behavior cloning of SFT, we propose to calibrate the preference critiques by utilizing the multi-agent feedback from downstream revision task, facilitating the robust RL for improving LLM's critique ability.

**Note that experiments in Section 6 verify the effectiveness of three solutions.** In conclusion, although our work builds upon existing critique and multi-agent research, it introduces novel technical solutions that significantly differ from previous works.

---

## **GPT-4 Dependency**

We appreciate the reviewer rDX7 and zQhf's concerns about GPT-4 dependency and novelty. We would like to clarify these points:

### **Flexible Framework**

**Our MultiCritique framework is a general and model-agnostic framework.** GPT-4 serves as one possible meta-critique judge model. While we chose GPT-4 due to its advanced meta-critique capabilities [1], any sufficiently advanced LLM can fulfill this role.  To demonstrate this flexibility, we conducted experiments with Claude-3.5-Sonnet and Qwen2.5-72B-Instruct. Specifically, we randomly sample 200 samples from MultiCritiqueDataset-SFT, and prompt Qwen2.5-72B-Instruct and Claude-3.5-Sonnet to re-run the Meta-Critique Classification and Critique Summarization processes. Subsequently, we assess two metrics to reflect the correlations between GPT-4 and these models: (1) Spearman correlation scores between GPT-4 and these models on assessing the response quality; (2) Agreement on judging the correctness of each ACU in Meta-Critique classification.

|  |Correlation| Agreement|
| - | - | - |
| Claude-3.5-Sonnet|0.58| 0.71|
| Qwen2.5-72B-Instruct |0.60|0.72|

These models achieve not only high meta-critique agreement (> 70%, the random baseline is 50%) with GPT-4 but also strong correlation
 (≈ 0.6) in final judgment scores. Therefore, other advanced models could also be used in our proposed MutiCritique.

> [1] The Critique of Critique

### **GPT-4-Free MultiCritique-RL**
More importantly, our MultiCritique-RL phase operates entirely without GPT-4, instead utilizing multiple 7B LLMs.

**In conclusion, while we leverage GPT-4's capabilities for practical implementation, any advanced models can fulfill this role.**

---

### Meta-Review · Area_Chair_2uRz · 2024-12-21

**Metareview:**

After carefully considering the six expert reviews and the subsequent author-reviewer discussions, I recommend rejecting this submission. While the paper presents an interesting approach to improving language models' critique abilities through multi-agent feedback, several concerns prevent it from meeting ICLR's acceptance standards.

The paper's primary technical contribution centers on MultiCritique, a data generation pipeline that leverages multiple language models to generate and refine critiques. While this approach demonstrates some improvements over existing methods, the reviewers raised significant concerns about both the novelty of the technical contribution and the cost-effectiveness of the proposed solution.

A major concern, highlighted particularly by reviewers fcKr and CtUU, is the substantial computational overhead introduced by the multi-agent feedback process and its heavy reliance on GPT-4. The authors' response does not fully address these efficiency concerns.

Several reviewers also questioned the methodological novelty of the work. As reviewer CtUU noted, the paper appears to build upon existing multi-agent debate and critique frameworks without introducing substantially new technical innovations. While the authors mentioned that their unified framework differs from previous work by addressing both SFT and RL phases, the core mechanisms still rely heavily on established techniques.

The discussion period revealed additional concerns about the method's scalability and generalizability. While the authors provided new experiments showing results with different base models and on unseen domains, questions remain about the approach's effectiveness with larger models and more diverse critique tasks. The authors' acknowledgment that improvements on 70B models were limited further suggests constraints on the method's broader applicability.

In weighing these factors, the combination of substantial computational overhead, limited technical novelty, and questions about broader applicability make this submission not ready for publication at ICLR 2025.

**Additional Comments On Reviewer Discussion:**

During the rebuttal period, there was extensive discussion between the authors and reviewers regarding several aspects of the paper. The discussions centered primarily on three main concerns: computational complexity and efficiency, technical novelty, and the method's broader applicability.

A significant concern raised by multiple reviewers focused on the computational overhead of the multi-agent system and its reliance on GPT-4. The authors responded by providing detailed cost analyses, demonstrating a 2.15-4.22x improvement in data efficiency compared to baselines. They argued that while their approach requires substantial computational resources during training, the improved data quality justifies this one-time investment. However, reviewers fcKr and CtUU remained unconvinced, noting that the performance improvements were modest relative to the computational costs.

The question of technical novelty generated substantial discussion. Several reviewers questioned whether the approach represented a significant advance over existing multi-agent frameworks. The authors responded by clarifying their contributions across three dimensions: introducing crucial information to reduce task complexity, aggregating diverse critiques from multiple agents, and calibrating preference critiques through downstream revision tasks. While some reviewers found these clarifications helpful, others maintained that the core mechanisms relied too heavily on established techniques.

The authors made commendable efforts to address concerns about broader applicability and generalization. They provided new experimental results showing performance on unseen domains and with different base models, including preliminary results with 70B models. However, their acknowledgment that improvements on larger models were limited, combined with questions about the framework's dependence on high-capability models for critique refinement, left some reviewers skeptical about the method's practical scalability.

In weighing these discussions for the final decision, I found the persistent concerns about computational efficiency particularly compelling. While the authors provided evidence of improved data efficiency, the substantial computational requirements and reliance on powerful models like GPT-4 raise questions about the method's broader adoption. Moreover, the limited technical novelty and questions about scalability to larger models suggest that more fundamental innovations may be needed to advance the field of automated critique generation.

These factors, combined with the authors' inability to fully address concerns about computational overhead despite their detailed responses, support the decision to reject the submission for ICLR 2025.

---

### Decision · Program_Chairs · 2025-01-22

Reject